# Asymmetric Quantum Multicast Network Coding: Asymmetric Optimal Cloning over Quantum Networks

Yuichi Hirota and Masaki Owari *

Department of Computer Science, Faculty of Informatics, Shizuoka University, Hamamatsu 432-8011, Japan; yu1.16ta@gmail.com
* Correspondence: masakiowari@inf.shizuoka.ac.jp

**Abstract:** Multicasting of quantum states is an essential feature of quantum internet. Since the noncloning theorem prohibits perfect cloning of an unknown quantum state, an appropriate protocol may depend on the purpose of the multicast. In this paper, we treat the multicasting of a single copy of an unknown state over a quantum network with free classical communication. We especially focus on protocols *exactly* multicasting an asymmetric optimal universal clone. Hence, these protocols are optimal and universal in terms of mean fidelity between input and output states, but the fidelities can depend on target nodes. Among these protocols, a protocol spending smaller communication resources is preferable. Here, we construct such a protocol attaining the min-cut of the network described as follows. Two (three) asymmetric optimal clones of an input state are created at a source node. Then, the state is divided into classical information and a compressed quantum state. The state is sent to two (three) target nodes using the quantum network coding. Finally, the asymmetric clones are reconstructed using LOCC with a small amount of entanglement shared among the target nodes and the classical information sent from the source node.

**Keywords:** quantum information; quantum communication; quantum network; universal cloning; network coding; entanglement

## 1. Introduction

In recent years, rapid progress has been made in the research and development of standalone quantum computers [1], both in terms of software and hardware, to the point where it is debated whether quantum computational supremacy has achieved [2–4]. In the near future, standalone quantum computers are expected to show innovative performance in various fields, such as machine learning [5–9] and computational chemistry [10–14]. On the other hand, it is known that much quantum information processing, including various different types of quantum cryptography such as quantum public-key cryptography [15], quantum blind computation [16,17], and quantum money [18,19] cannot be realized on standalone quantum computers but only on quantum networks [20], where a quantum network of a large size is called quantum internet [21,22]. Thus, recently, quantum network has been intensively studied both theoretically [23–29] and experimentally [30,31].

Theoretical research for improving the throughput of quantum networks started from the study of quantum repeaters aiming to share a maximally entangled state between end vertices of a quantum network represented by a path graph [32,33]. After an enormous amount of research in this direction (see [20] and references therein), the ultimate limit for sharing a maximally entangled state between a given pair of nodes on a quantum network was finally derived by Pirandola et al. [24,26]. On the other hand, since multiple users may simultaneously communicate with each other, it is also important to study multiparty communication protocol on a quantum network. In general, the problem of finding a better multiparty communication protocol on a quantum network is much more complicated than a problem of a two-party communication protocol. First, even on the classical network, there are many different types of multiparty communication. Simple examples of

multiparty communication may be multiple-unicast communication and multicast communication, where the definitions of these schemes are given later in this introduction. However, in general, an arbitrary type of communication among multiple parties can be considered [34], and we need to optimize the communication protocol depending on the type. Furthermore, there is an additional problem with multiparty communication on quantum networks. As is well known, "a class of maximally entangled states" is not unique [35] in a multipartite system, which is represented by the incomparability of GHZ states and W states under stochastic local operation and classical communication (SLOCC) [36]. This fact suggests that depending on our purpose, we need to use a different multiparty protocol to share a different type of states. In other words, we cannot discuss the optimality of the multiparty protocol before we determine what type of states we want to share.

In classical information theory, a technique called network coding is known to be useful to improve throughput of various different multiparty communication schemes when there is a bottleneck on a network. Here, network coding is a technique of applying nontrivial operations to the bitstream at intermediate nodes [37–39]. The method of network coding can be also applied to quantum networks, and quantum information processing on a quantum network that utilizes methods of network coding is called "a quantum network coding" [40]. There has been a considerable amount of research on quantum network coding, which tries to improve the throughput of a quantum network in various situations [41–57]. Recently, it has been presented that quantum network coding can improve the security of a quantum network [58–61] and reduce decoherence effect [62]. Furthermore, it is useful for quantum repeater networks [63,64] as well as for distributed quantum computation [65]. Moreover, a simple quantum network code has been experimentally demonstrated [66,67]. Although many studies have considered network coding on noisy classical networks in classical information theory, almost all the studies of quantum network coding consider noise-free quantum networks. This is because quantum network coding is regarded as a protocol implemented on a layer on which the errors have been already corrected. Hence, in this study, we consider noise-free quantum networks.

In classical network coding, the majority of the studies have focused on multicast communication, where a single source node transmits the same information to multiple target nodes on a network [37,38]. The left-hand side of Figure 1 shows the network coding for a the butterfly network. This is one of the simplest examples of classical multicast network coding. Another type of network coding is called multiple-unicast network coding. Here, there are $k$ pairs of source and target nodes $(s_0, t_0), \ldots, (s_{k-1}, t_{k-1})$ on the network, and each source node $s_i$ independently transmits a message to the corresponding target node $t_i$ for all $i$ [68]. The modified version of the butterfly network in the right-hand side of Figure 1 is one of the simplest examples of classical multiple-unicast network coding.

Most of the research on quantum network coding considered multiple-unicast communication, i.e., multiple-unicast quantum network coding, where each source node transmits a quantum state (instead of a classical message) to the corresponding target node [41,43,45,46,53,58–61,63,64]. The most important results are those of Kobayashi et al. If classical information (or measurement results) can be freely sent among the nodes on a quantum network, Kobayashi et al. gave a canonical procedure for constructing a quantum multiple-unicast network code from a given classical multiple-unicast network code. Here, the quantum network for the quantum code and the classical network for the classical code must be represented by the same graph [43,46].

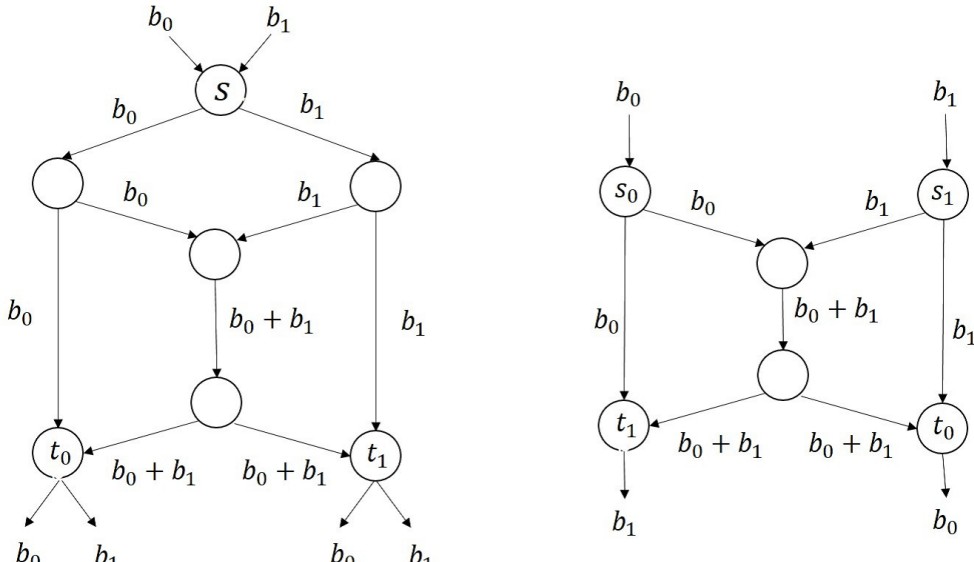

**Figure 1.** The left-hand side is a multicast classical network coding on the butterfly network, where a single source node $s$ sends messages $b_0$ and $b_1$ on the finite field $\mathbb{F}_q := \mathbb{Z}/\mathbb{Z}_q$ to both target nodes $t_1$ and $t_2$, where $q$ is a prime power. The right-hand side is a multiple-unicast classical network coding on the butterfly network, where a source node $s_0$ sends message $b_0 \in \mathbb{F}_q$ to target node $t_0$, and source node $s_1$ sends message $b_1 \in \mathbb{F}_q$ to target node $t_1$.

Unlike quantum multiple-unicast network coding, there has been less research on quantum network coding focusing on multicast communication [42,44,48–50,52,57]. This is because in quantum information theory, the no-cloning theorem prohibits perfect multicast communication [69], and, thus, it is not straightforward to construct a multicast quantum network coding protocol as an extension of a classical multicast network coding protocol.

One paper by Shi et al. is the first to treat quantum multicast network coding [42]. They consider the problem of distributing $N$-identical copies of a state $|\psi\rangle$ from a single source node to $N$ target nodes. Since the number of copies of $|\psi\rangle$ is equal to the number of target nodes, $|\psi\rangle$ can be distributed without cloning the quantum states. Shi et al. showed that coding on intermediate nodes can increase the throughput of the quantum network.

The second work treating this topic is one paper by Kobayashi et al. [44]. In this paper, a single copy of a state $|\psi\rangle = \sum_{i=1}^{d} \alpha_i |i\rangle$ is given on the source node and the aim is to share a Greenberger–Horne–Zeilinger (GHZ)-type state $\sum_{i=1}^{d} \alpha_i |i\rangle_1 \otimes \cdots \otimes |i\rangle_N$ among target nodes, where the $i$th local system is on the $i$th target node. From this GHZ-type state shared among the target nodes, the input state $|\Psi\rangle$ can be reconstructed at any target node by local operations and classical communication (LOCC). Based on classical multicast network coding, Kobayashi et al. developed a quantum protocol to achieve the above task under the assumption of free classical communication among nodes on the quantum network.

The third work treating this topic is one paper by Xu et al. [52]. They proposed a communication protocol on a quantum network called quantum cooperative multicast. In their problem setting, there are multiple source nodes $s_1, \cdots, s_N$ and target nodes $t_1, \cdots, t_M$ in a given quantum network. At the beginning, source node $s_i$ has an unknown state $|\psi_i\rangle = \sum_{k=1}^{d} \alpha_{i,k} |k\rangle$, and their purpose is that for a given function $f_k : \mathbb{C}^N \to \mathbb{C}$, after quantum communication over the quantum network, each target node reconstructs a quantum state $|\phi\rangle$ defined by

$$|\phi\rangle = \sum_{k=1}^{d} f_k(\alpha_{1,k}, \cdots, \alpha_{N,k}) |k\rangle. \tag{1}$$

They showed that this problem can be considered as a generalization of classical multicast communication and presented protocols using network coding to probabilistically achieve the above goal on the multiple-unicast butterfly network given in the right figure of Figure 1. Although the protocols of Shi et al., Kobayashi et al., and Xu et al. can be considered as generalizations of classical multicast network coding to quantum networks, rigorously speaking, the goal of their protocols is not exactly to achieve a multicast of a quantum state.

Recently, Pan et al. proposed a new multicast protocol using quantum network coding [57]. The purpose of the protocol is to probabilistically send an exact copy or an exact orthogonal complement of a known qubit state from a source node to multiple target nodes on a quantum network. Because the sender knows the state, this protocol does not contradict the no-cloning theorem [69]. On the other hand, this task is trivial when free classical communication is allowed. Hence, they consider the situation where each channel on the quantum network can send either one qubit or two classical bits in a single session of the protocol; that is, classical communication is restricted in their problem setting. They give an efficient protocol to achieve this goal on the multicast butterfly network given in the left figure of Figure 1 and also on the extended butterfly network.

In this paper, we consider a yet different problem setting on multicast on the quantum network. As we have already mentioned, perfect multicast of an unknown state is impossible. Nevertheless, imperfect multicast of an unknown quantum state is still possible. When we restrict ourselves to imperfect multicast of a quantum state through noiseless quantum channels, the problem of multicast of an unknown quantum state reduces to a problem of cloning an unknown quantum state. This problem is called quantum approximate cloning [70,71] and has been intensively studied both theoretically [72–85] and experimentally [86–91].

The performance of a cloning protocol is normally measured in terms of the fidelity between an unknown input state and an output clone. When this fidelity does not depend on an input state, the protocol is called universal cloning. Among them, a protocol achieving the maximum fidelity is called optimal universal cloning [70,71]. Here, we need to add the following remark. When $M$ output clones $\rho_i$ ($i = 1 \cdots M$) of an input unknown state, $|\psi\rangle$ are created by a cloning protocol. We need to treat $M$ independent fidelities $F_i := \langle \psi | \rho_i | \psi \rangle$ and cannot straightforwardly define an optimal protocol. One way to resolve this problem is to add a constraint that all the clones are equivalent, that is, $\rho_i = \rho_j$ for all $i$ and $j$. This immediately leads $F_i = F_j$ for all $i$ and $j$, and the optimal universal cloning satisfying this condition is called symmetric optimal universal quantum cloning. On the other hand, when we do not use the constraint of symmetry, we need to define a weight $w_i$ satisfying $w_i \geq 0$ and $\sum_i w_i = 1$ and a weighted mean fidelity $\overline{F} := \sum_i w_i F_i$. The universal cloning which is optimal in terms of $\overline{F}$ is called asymmetric optimal universal quantum cloning. By the definition, symmetric optimal universal cloning is a special case of asymmetric optimal universal quantum cloning such that the weight $w_i = 1/M$ for all $i$.

In the viewpoint of multiparty communication, optimal universal quantum cloning is nothing but a multicast quantum channel that is optimal in terms of fidelity. Therefore, when we consider a quantum network communication, sending an optimal universal quantum clone (UQC) from a single source node to multiple target nodes is a multicast network communication that is optimal in terms of fidelity. Furthermore, this problem may be considered one of the most natural quantum extensions of classical multicast network coding. Hence, our research problem to study in which condition this communication task is achievable is information-theoretically important. Furthermore, it is known that quantum approximate cloning is useful for several information processing tasks including eavesdropping on quantum key distribution protocols [92–96], broadcasting quantum coherence [97], and multiple-unicast quantum network coding over the butterfly network without classical communication [40]. This fact strongly suggests that our research problem is practically important as well.

Based on this idea, Owari et al. constructed a protocol to exactly share a symmetric optimal UQC of an input state on the target nodes under the conditions that classical information can be sent freely among nodes on a quantum network and that a small amount of entanglement is shared on target nodes at the beginning of the protocol [49,50]. They also gave a protocol to approximately share a symmetric optimal UQC without entanglement shared among target nodes [48]. We further note that although the references [48–50] are written in Japanese, their English version is now planned to be written [98].

In this paper, we focus on extending Owari et al.'s results [49,50] to asymmetric optimal universal quantum cloning [75–85], which is a generalization of symmetric optimal universal quantum cloning. Thus, we construct a protocol to efficiently and *exactly* multicast an asymmetric optimal clone of a $q^r$-dimensional input quantum state from one source node to two (three) target nodes, *where q is assumed to be a prime power.*

Our problem setting is given by the following four assumptions, which are almost in common with those used in [49,50]:

- The noise-free quantum network can be described by an undirected graph $G$ with one source node and two (three) target nodes.
- Each quantum channel on the quantum network can transmit one $q$-dimensional quantum system in a single session.
- Measurement results (or classical information) can be sent freely from one node to another node on the quantum network.
- A small amount of entanglement that does not scale with $q$, is shared among the target nodes. The amount of entanglement is at most $2$ ebit for two target nodes, and at most $(2 + 4\log_2 3)$ ebit for the case of three target nodes.

Under these assumptions, we prove that multicasting of $1 \rightarrow 2$ ($1 \rightarrow 3$) asymmetric optimal UQCs of a $q^r$-dimensional state is possible, if there exists a classical solvable linear multicast network code with source rate $r$ for a noise-free classical network described by an acyclic directed graph $G'$, where $G$ is an undirected underlying graph of $G'$. Using the max-flow and min-cut theorem of multicast network coding [37,38], for sufficiently large $q$, this sufficient condition for the existence of a classical network code on $G'$ can be replaced by the condition that the minimum-cut between the source node $s$ and a target node $t_i$ is no less than $r$ for all $i$.

An outline of our protocol is as follows:

- We create two (three) asymmetric optimal UQCs of an input state with an ancilla system at a source node.
- We measure the ancilla system and send the measurement outcomes to the target nodes.
- We compress the whole $d^2$ ($d^3$)-dimensional system into a $d$-dimensional system.
- We transmit the resulting state to two (three) target nodes using Kobayashi et al.'s multicast quantum network coding [44]. As a result, a GHZ-type state is shared among target nodes.
- We reconstruct the asymmetric optimal UQCs of the input state from the GHZ-type state using LOCC with a small amount of entanglement among the target nodes and the measurement outcomes sent from the source node.

Using the above protocol, we can multicast asymmetric optimal clones from one source node to two (three) target nodes (Figure 2). Here, we note that although the above outline of our protocol is almost the same as the protocol of Owari et al. of symmetric optimal UQCs, the detail of each part is completely different.

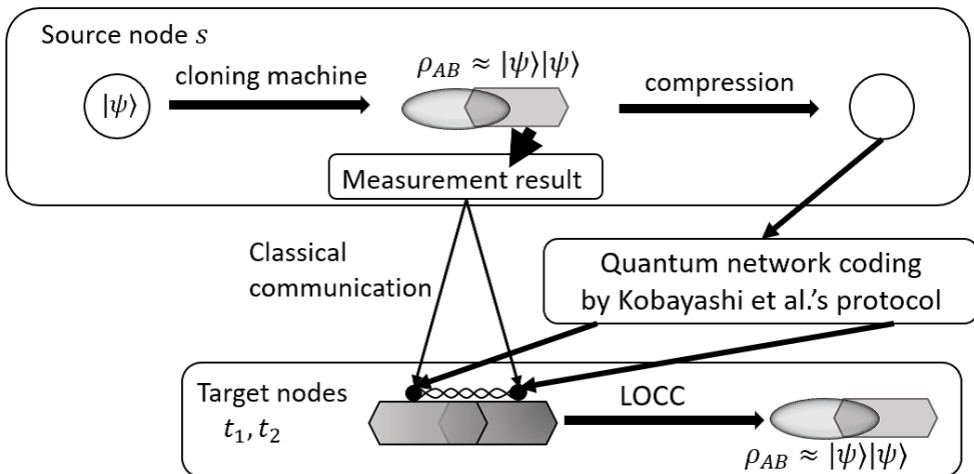

**Figure 2.** Schematic diagram of a protocol for multicasting asymmetric optimal UQCs from one source node to two target nodes. The asymmetric optimal cloning protocol for the input state $|\psi\rangle$ is implemented at the source node. The resulting state is compressed into a $d$-dimensional system and transmitted to the two target nodes using the quantum multicast network coding protocol of Kobayashi et al. [44]. Finally, the asymmetric optimal clones of the input state are reconstructed by LOCC on target nodes with the help of a small amount of entanglement.

To see the efficiency of our protocol, it is convenient to consider the multicast butterfly network given on the left-hand side of Figure 1. When each edge corresponds to a $q$-dimensional noiseless quantum channel and the target nodes $t_0$ and $t_1$ share 2 ebit, our protocol distributes asymmetric optimal QCMs of $q^2$-dimensional input states. On the other hand, we can easily see that a conventional protocol without network coding based on entanglement swapping (or quantum repeater) only distributes asymmetric optimal QCMs of $q$-dimensional input states at most. Hence, the rate of our protocol is double that of the conventional protocol. We further show that the above protocol can be used for efficient preparation of quantum asymmetric telecloning [99,100] over a quantum network.

The rest of the paper is organized as follows: We explain the asymmetric optimal universal quantum cloning and the quantum multicast network coding protocol of Kobayashi et al. in Section 2. We present all the results in Section 3, where a protocol for multicasting $1 \to 2$ asymmetric optimal clones is given in Section 3.1. We also present a protocol for multicasting $1 \to 3$ asymmetric optimal clones in Section 3.2. Finally, we give a discussion and summary in Section 4.

## 2. Materials and Methods

Asymmetric optimal universal quantum cloning, classical linear multicast network coding, and the multicast quantum network coding protocol of Kobayashi et al. are all important in our protocol. In this section, we explain optimal asymmetric universal quantum cloning in Section 2.1. Then, classical linear multicast network coding and the multicast quantum network coding protocol of Kobayashi et al. are presented in Sections 2.2 and 2.3, respectively.

### 2.1. Optimal Asymmetric Quantum Universal Cloning Machine

The no-cloning theorem states that quantum mechanics prohibits a quantum operation that makes perfect copies of an unknown quantum state [69]. That is, there is no quantum channel (or a completely positive and trace preserving map) $\varepsilon : \mathfrak{B}(\mathcal{H}) \to \mathfrak{B}(\mathcal{H}^{\otimes 2})$ satisfying $\varepsilon(\rho) = \rho \otimes \rho$ for all pure states $\rho$ on a Hilbert space $\mathcal{H}$, where $\mathfrak{B}(\mathcal{H})$ is a space of all linear operators on the Hilbert space $\mathcal{H}$. This immediately leads to the impossibility of a perfect multicast of an unknown quantum state. On the other hand, quantum mechanics does not completely prohibit approximate cloning of a quantum state. Thus, there may exist a quantum channel $\varepsilon : \mathfrak{B}(\mathcal{H}) \to \mathfrak{B}(\mathcal{H}^{\otimes 2})$ such that $\mathrm{Tr}_2\, \varepsilon(\rho)$ and $\mathrm{Tr}_1\, \varepsilon(\rho)$ are close to $\rho$ for

an arbitrary pure state $\rho$, where $\mathrm{Tr}_i$ is the partial trace of the $i$th subsystem. Hence, many studies have focused on quantum protocols to make an approximate copy of unknown states (so-called quantum cloning machines) [72–85].

In general, a quantum cloning machine (QCM) that produces $N$ approximate clones based on $M$ copies of a given quantum state $|\psi\rangle \in \mathcal{H}$ is a quantum channel $\varepsilon$ from $\mathfrak{B}(\mathcal{H}^{\otimes M})$ to $\mathfrak{B}(\mathcal{H}^{\otimes N})$. Suppose that $\rho_i$ is a reduced density matrix of the output state on the $i$th subsystem: $\rho_i = \mathrm{Tr}_{\neg i}\,\varepsilon\big((|\psi\rangle\langle\psi|)^{\otimes M}\big)$, where $\mathrm{Tr}_{\neg i}$ is a partial trace of all subsystems except the $i$th subsystem. Since the purpose of a QCM is to make $\rho_i$ as closed as the input state $|\psi\rangle\langle\psi|$, the performance of a QCM can be described by the output fidelity $F_i$ between $\rho_i$ and $|\psi\rangle\langle\psi|$:

$$F_i = \langle\psi|\rho_i|\psi\rangle, \quad (i = 1, ..., M). \tag{2}$$

A QCM is called universal if $F_i$ does not depend on the input state $|\psi\rangle$. Furthermore, a universal QCM (UQCM) is called symmetric if the all clones are the same: $\rho_i = \rho_j$ for all $i$ and $j$. A UQCM that is not symmetric is asymmetric. Since the output states of an asymmetric UQCM satisfy $\rho_i \neq \rho_j$, the output fidelity $F_i$ also depends on $i$. Thus, to discuss the optimality of an asymmetric UQCM, we need to define a weight $\{w_i\}_{i=1}^{M}$ satisfying $w_i \geq 0$ and $\sum_i^M w_i = 1$. Then, the mean output fidelity $\overline{F}$ of an asymmetric UQCM $\varepsilon$ with respect to the weight $\{w_i\}_{i=1}^{M}$ is defined by

$$\overline{F} := \sum_i^M w_i F_i. \tag{3}$$

For a given weight $\{w_i\}_{i=1}^{M}$, an asymmetric UQCM that attains the optimum of the mean output fidelity $\overline{F}$ is called an asymmetric optimal UQCM w.r.t $\{w_i\}$.

For existing results on asymmetric optimal UQCM, we refer to [71,84]. Here, we only give known facts on asymmetric optimal UQCM that are necessary for our research. First, we give an optimal asymmetric UQCM with $M = 1$ and $N = 2$ (we call this protocol a $1 \to 2$ optimal asymmetric UQCM). This protocol uses three systems: $A$, $B$, and $M$, whose Hilbert spaces are $\mathcal{H}_A$, $\mathcal{H}_B$, and $\mathcal{H}_M$, respectively. Here, $\mathcal{H}_A$ works as an input system and the first output system, $\mathcal{H}_B$ is the second output system, and $\mathcal{H}_M$ is an ancilla system. The dimensions of all three systems are the same, and we denote this dimension as $d$; that is, $\dim \mathcal{H}_A = \dim \mathcal{H}_B = \dim \mathcal{H}_M =: d$. Then, for an input state $|\psi\rangle$ on system $A$, a $1 \to 2$ optimal asymmetric UQCM is given by an isometry $U_{1\to2}^{(a,b)}$ from $\mathcal{H}_A$ to $\mathcal{H}_A \otimes \mathcal{H}_B \otimes \mathcal{H}_M$ satisfying [78]:

$$U_{1\to2}^{(a,b)}|\psi\rangle_A = a|\psi\rangle_A|\Phi_d^+\rangle_{BM} + b|\psi\rangle_B|\Phi_d^+\rangle_{AM}. \tag{4}$$

where $|\Phi_d^+\rangle$ is a standard $d$-dimensional maximally entangled state:

$$|\Phi_d^+\rangle := \frac{1}{\sqrt{d}}\sum_{k=0}^{d-1}|k\rangle|k\rangle, \tag{5}$$

and $a$ and $b$ are positive real parameters satisfying

$$a^2 + b^2 + \frac{2ab}{d} = 1. \tag{6}$$

Note that in order to simplify the formulas, the parameters $a, b$ are often used instead of the weight $\{w_i\}_{i=1}^{2}$ in the definition of the $1 \to 2$ optimal asymmetric UQCM. Using $U_{1\to2}^{(a,b)}$ defined above, the optimal asymmetric UQCM $\varepsilon_{1\to2}^{(a,b)}$ is

$$\varepsilon_{1\to2}^{(a,b)}(|\psi\rangle\langle\psi|) := \mathrm{Tr}_M\Big(U_{1\to2}|\psi\rangle\langle\psi|_A U_{1\to2}^{\dagger}\Big). \tag{7}$$

The reduced density matrices of the output states can be written as

$$\rho_A := \text{Tr}_B \, \varepsilon_{1 \to 2}^{(a,b)}(|\psi\rangle\langle\psi|) = (1 - b^2)|\psi\rangle\langle\psi| + b^2 \frac{I}{d}$$
$$\rho_B := \text{Tr}_A \, \varepsilon_{1 \to 2}^{(a,b)}(|\psi\rangle\langle\psi|) = (1 - a^2)|\psi\rangle\langle\psi| + a^2 \frac{I}{d}. \tag{8}$$

Thus, the fidelity of the reduced density matrices, which have been proved to be optimum [78], is given by

$$F_A := \langle\psi|\text{Tr}_B\left(\varepsilon_{1 \to 2}^{(a,b)}(|\psi\rangle\langle\psi|)\right)|\psi\rangle = 1 - b^2 \frac{d - 1}{d},$$
$$F_B := \langle\psi|\text{Tr}_A\left(\varepsilon_{1 \to 2}^{(a,b)}(|\psi\rangle\langle\psi|)\right)|\psi\rangle = 1 - a^2 \frac{d - 1}{d}, \tag{9}$$

where $I$ is an identity operator on a $d$-dimensional system. The well-known formula $F_A = F_B = \frac{d+3}{2(d+1)}$ of the fidelity of an optimal symmetric UQCM is derived from the above equations by substituting $a = b$ [71].

Next, we give an optimal asymmetric UQCM with $M = 1$ and $N = 3$ (we call this protocol the $1 \to 3$ optimal asymmetric UQCM). This protocol use five systems $A$, $B$, $C$, $R$, and $S$ whose Hilbert spaces are $\mathcal{H}_A$, $\mathcal{H}_B$, $\mathcal{H}_C$, $\mathcal{H}_R$, and $\mathcal{H}_S$, respectively. Here, $\mathcal{H}_A$ is an input system that is also the first output system. $\mathcal{H}_B$ and $\mathcal{H}_C$ are the second and third output systems, respectively. $\mathcal{H}_R$ and $\mathcal{H}_S$ are ancilla systems. The dimensions of all systems are the same, which we denote as $d$. For an input state $|\psi\rangle$ on system $A$, $1 \to 3$ optimal asymmetric UQCM is given by an isometry $U_{1 \to 3}^{(\alpha,\beta,\gamma)}$ from $\mathcal{H}_A$ to $\mathcal{H}_A \otimes \mathcal{H}_B \otimes \mathcal{H}_C \otimes \mathcal{H}_R \otimes \mathcal{H}_S$ satisfying the following equation:

$$U_{1 \to 3}^{(\alpha,\beta,\gamma)}|\psi\rangle = \sqrt{\frac{d}{2d + 2}}[\alpha|\psi\rangle_A(|\Phi_d^+\rangle_{BR}|\Phi_d^+\rangle_{CS} + |\Phi_d^+\rangle_{BS}|\Phi_d^+\rangle_{CR})$$
$$+ \beta|\psi\rangle_B(|\Phi_d^+\rangle_{AR}|\Phi_d^+\rangle_{CS} + |\Phi_d^+\rangle_{AS}|\Phi_d^+\rangle_{CR})$$
$$+ \gamma|\psi\rangle_C(|\Phi_d^+\rangle_{AR}|\Phi_d^+\rangle_{BS} + |\Phi_d^+\rangle_{AS}|\Phi_d^+\rangle_{BR})]. \tag{10}$$

Here, $\alpha, \beta, \gamma$ are non-negative real parameters which are used instead of the weight $\{w_i\}_{i=1}^3$ and satisfy the following constraint [71,79,80]:

$$\alpha^2 + \beta^2 + \gamma^2 + \frac{2}{d}(\alpha\beta + \beta\gamma + \alpha\gamma) = 1. \tag{11}$$

In terms of $U_{ABCRS}$, a $1 \to 3$ optimal asymmetric UQCM $\varepsilon_{1 \to 3}^{(\alpha,\beta,\gamma)}$ can be written as:

$$\varepsilon_{1 \to 3}^{(\alpha,\beta,\gamma)}(|\psi\rangle\langle\psi|) := \text{Tr}_{RS}\left(U_{1 \to 3}^{(\alpha,\beta,\gamma)}|\psi\rangle\langle\psi|_A U_{1 \to 3}^{(\alpha,\beta,\gamma)\,\dagger}\right). \tag{12}$$

The fidelities between an input state and each reduced density matrix, which were proved to be optimum [71,79,80], are given as follows:

$$F_A = 1 - \frac{d - 1}{d}\left(\beta^2 + \gamma^2 + \frac{2\beta\gamma}{d + 1}\right),$$
$$F_B = 1 - \frac{d - 1}{d}\left(\alpha^2 + \gamma^2 + \frac{2\alpha\gamma}{d + 1}\right),$$
$$F_C = 1 - \frac{d - 1}{d}\left(\alpha^2 + \beta^2 + \frac{2\alpha\beta}{d + 1}\right). \tag{13}$$

We refer [79] for the figure depicting behavior of $F_A$, $F_B$, and $F_C$ with respect to $\alpha, \beta$ and $\gamma$.

### 2.2. Classical Multicast Network Coding

Since our protocol uses the protocol of Kobayashi et al. as a subroutine and since the protocol of Kobayashi et al. is based on a classical linear multicast network code, we introduce classical linear multicast network coding in this subsection. A detailed description of classical multicast network coding can be found in standard textbooks of network coding [37,38].

A classical network is represented by a directed graph $G' = (V, E')$, where a vertex $v \in V$ represents a node of the network and an edge $e \in E'$ represents a noiseless classical channel. In this paper, we assume that $G'$ is *acyclic*. There exist a source node $s \in V$ and $N$ target nodes $t_1, \ldots, t_N \in V$ on the network. A node that is neither a source node nor a target node is called an intermediate node. In a single session of a classical multicast network coding, an alphabet on the finite field $\mathbb{F}_q$ is sent from node $u$ to node $v$ if $(u, v) \in E'$, where the order of $\mathbb{F}_q$ is a prime power $q$. Since $G'$ is an acyclic directed graph, a natural partial ordering can be defined on $E'$. This partial ordering is generated by the condition that $(u, v), (v, w) \in E' \implies (u, v) \prec (v, w)$. The order of transmissions of classical information can be determined by this partial ordering. That is, an edge $e \in E'$ transmits an alphabet after all edges $e' \in E'$ satisfying $e' \prec e$ have transmitted alphabets. We assume that there is no incoming edge to the source node $s$ and that there is no outgoing edge from any target node. Hence, all edges whose tail node is the source node $s$ are a local minimum, and all edges whose head node is a target node are a local maximum under the partial ordering. We further assume that all edges whose tail node is not the source node $s$ are not a local minimum and that all edges whose head node is not a target node are not a local maximum. This is because the edges that do not satisfy these conditions are useless for our purpose.

A classical linear multicast network code over $\mathbb{F}_q$ on $G'$ consists of a set of linear maps $\{f_e\}_{e \in E'}$. At the beginning of a session, an input message $\vec{x} := (x_1, \ldots, x_r) \in \mathbb{F}_q^r$ is chosen on the source node $s$, where $r$ is the source rate of the classical multicast network code. Suppose $e$ is an outgoing edge of $v$. At the first step of the network coding, an alphabet $y_e$ transmitted through the edge $e$ is chosen as a linear combination of $x_1, \ldots, x_r$. In other words, in terms of a linear function $f_e : \mathbb{F}_q^r \to \mathbb{F}_q$, $y_e$ can be written as

$$y_e := f_e(\vec{x}) = f_e(x_1, \cdots x_r). \tag{14}$$

After calculating $y_e$, $y_e$ is transmitted through $e$. After all edges outgoing from the source node $s$ transmitted an alphabet, all intermediate nodes transmit alphabet in the order determined by the partial ordering as follows: Suppose an intermediate node $v$ on the network has $m$ incoming edges and $e$ is an outgoing edge from $v$. After all transmissions of $m$ incoming edges to $v$ have finished, the node $v$ has $m$-alphabets $y_j \in \mathbb{F}_q$ ($j = 1, \ldots, m$), where $y_j$ is an alphabet sent through the $j$th incoming edge. Then, an alphabet $y_e$ transmitted through the edge $e$ is chosen as a linear combination of $y_1, \ldots, y_m$. In other words, there exists a linear function $f_e : \mathbb{F}_q^m \to \mathbb{F}_d$ such that

$$y_e := f_e(y_1, \cdots y_m). \tag{15}$$

After the calculation, $y_e$ is transmitted through $e$.

Suppose a target node $t_i$ has $m_i$ incoming edges. Then, after all edges have transmitted an alphabet, the target node $t_i$ has $m_i$-alphabets $y_j^{(i)} \in \mathbb{F}_q$ ($j = 1, \cdots, m_i$), where $y_j^{(i)}$ is an alphabet sent through the $j$th incoming edge to $t_i$. A classical linear multicast network code $\{f_e\}_{e \in E'}$ is called *solvable* if there exists a set of decoding operations $\{g_i\}_{i=1}^N$ such that $g_i : \mathbb{F}_q^{m_i} \to \mathbb{F}_q^r$ satisfies the following equation for all $i$:

$$\vec{x} = g_i\left(y_1^{(i)}, \cdots, y_{m_i}^{(i)}\right), \tag{16}$$

where $\vec{x} \in \mathbb{F}_q^r$ is the input message. If a classical linear multicast network code is solvable, any decoding operation $g_i$ can be chosen as a linear map.

There is a necessary and sufficient condition for the existence of a classical linear multicast network code [37,38]. Suppose that $C_i$ is the size of the minimum cut between $s$ and $t_i$. Then, there exists a classical linear multicast code with source rate $r$ on $G'$ over a sufficiently large field $\mathbb{F}_q$ if and only if $C_i \geq r$ for all $i$. This is nothing but a generalization of the famous max-flow min-cut theorem to a multicast network communication.

### 2.3. Quantum Multicast Network Coding

In this subsection, we review the protocol of Kobayashi et al. [44]. First, we give a problem setting for multicast quantum network coding that is common between our protocol and the protocol of Kobayashi et al. A quantum network is described by an *undirected* graph $G = (V, E)$, where $V$ represents a set of nodes and $E$ represents a set of quantum channels. There exist a source node $s \in V$ and $N$ target nodes $t_1, \ldots, t_N \in V$ on the network. In a single session, any quantum channel $(u, v) \in E$ can send a $q$-dimensional quantum system $\mathcal{H}_e$ just once either from $u$ to $v$, or from $v$ to $u$, where $q$ is assumed to be a prime power. Furthermore, any quantum operations can be implemented on any node $v \in V$, and measurement outcomes (or classical information) can be freely sent among nodes. At the beginning of a session, a single copy of input state $|\psi\rangle$ is given on the source node $s$. Here, the reason a quantum channel is represented by an undirected edge is that the direction of a quantum channel can be effectively reversed by quantum teleportation under the condition of free classical communication [45].

The purpose of both protocols is to multicast an input state $|\psi\rangle$ from the source node to all target nodes in a single session. Here, we should note that the meaning of "multicast" in the protocol of Kobayashi et al. is different from that in our protocol. As we have explained in the introduction, the purpose of our protocol is to construct optimal asymmetric universal clones among target nodes for a given $q^r$-dimensional input state $|\psi\rangle = \sum_{j=0}^{q^r-1} \alpha_j |j\rangle \in \mathcal{H}_s$ on a source node, where $\mathcal{H}_s$ is a $q^r$-dimensional input space. In other words, we consider multicast quantum network coding with source rate $r$. On the other hand, the purpose of the protocol of Kobayashi et al. is to construct a GHZ-type state $\sum_{j=0}^{q^r-1} \alpha_j |j\rangle_1 \otimes \cdots \otimes |j\rangle_N$ among target nodes, where the $i$th local system is on the $i$th target node.

Both the protocol of Kobayashi et al. and our protocol are constructed under the assumption that there exists a solvable classical linear multicast network code $\{f_e\}_{e \in E'}$ with source rate $r$ on an acyclic directed graph $G' = (V, E')$ over a finite field $\mathbb{F}_q$, where $G$ is an undirected underlying graph of $G'$. In other words, $G$ can be derived by replacing all directed edges on $G'$ by undirected edges. Using this replacement, a directed edge $e' \in E'$ is naturally mapped to an undirected $e \in E$, and this map is a bijection. Hence, in the following part of this subsection, we will not distinguish $e'$ from $e$ and will write $e'$ as $e$.

The protocol of Kobayashi et al. imitates a classical linear multicast network code $\{f_e\}_{e \in E'}$ and corresponding decoding operations $\{g_i\}_{i=1}^N$ by unitary operators. Each linear map $f_e$ is imitated by a unitary operator $U_e$, and each recovery operator $g_i$ is imitated by a unitary operator $V_i$, where $U_e$ and $V_i$ are defined as follows: Since $\dim \mathcal{H}_s = q^r$, due to the bijection between $\{0, 1 \ldots q^r - 1\}$ and $\mathbb{F}_q^r$, an input state $|\psi\rangle \in \mathcal{H}_s$ can be written as $|\psi\rangle = \sum_{\vec{x} \in \mathbb{F}_q^r} \alpha_{\vec{x}} |\vec{x}\rangle$. For an outgoing edge $e$ from the source node $s$, a unitary operator $U_e$ on $\mathcal{H}_s \otimes \mathcal{H}_e$ is defined by means of $f_e : \mathbb{F}_q^r \to \mathbb{F}_q$ as

$$U_e := \sum_{\vec{x} \in \mathbb{F}_q^r, y \in \mathbb{F}_q} |\vec{x}\rangle \langle \vec{x}|_s \otimes |y + f_e(\vec{x})\rangle \langle y|_{e'} \tag{17}$$

where $\mathcal{H}_e$ is a Hilbert space transmitted through $e$. Suppose $In(e)$ is a set of all incoming edges of $v$, where $v$ is a tail node of $e$, and suppose $\mathcal{H}_{In(e)} := \bigotimes_{e' \in In(e)} \mathcal{H}_{e'}$. Then, for an outgoing edge $e$ from an intermediate node $v$, a unitary operator $U_e$ on $\mathcal{H}_{In(e)} \otimes \mathcal{H}_e$ is defined by means of $f_e : \mathbb{F}_q^{|In(e)|} \to \mathbb{F}_q$ as

$$U_e := \sum_{\vec{y} \in \mathbb{F}_q^{|In(e)|}, y_e \in \mathbb{F}_q} |\vec{y}\rangle \langle \vec{y}|_{In(e)} \otimes |y_e + f_e(\vec{y})\rangle \langle y_e|_e. \tag{18}$$

Suppose $\mathcal{V}_i$ is a $q^r$-dimensional output Hilbert space on a target node $t_i$. A unitary operator $V_i$ on $\mathcal{H}_{In(t_i)} \otimes \mathcal{V}_i$ is defined by means of the decoding operation $g_i : \mathbb{F}_q^{|In(t_i)|} \to \mathbb{F}_q^r$ as

$$V_i := \sum_{\vec{y} \in \mathbb{F}_q^{|In(t_i)|}, \vec{x} \in \mathbb{F}_q^r} |\vec{y}\rangle\langle\vec{y}|_{In(t_i)} \otimes |\vec{x} + g_i(\vec{y})\rangle\langle\vec{x}|_i. \tag{19}$$

The quantum multicast network coding protocol of Kobayashi et al. is shown in Protocol 1.

---

**Protocol 1** The quantum multicast network coding protocol of Kobayashi et al.

---

**Step 1: Initialization**
At the beginning, the source node $s$ has an unknown initial state $|\psi\rangle$ on $\mathcal{H}_s$. Each node $v \in V$ prepares $|0\rangle$ on $\mathcal{H}_e$ for an edge $e \in E$ whose tail node is $v$. For all $i$ satisfying $1 \le i \le m$, a target node $t_i$ prepares $|0\rangle$ on $\mathcal{V}_i$.
**Step 2: Transmission**
First, for all edges $e \in E'$ whose tail node is the source node, the source node operates the unitary operator $U_e$ on $\mathcal{H}_s \otimes \mathcal{H}_e$ and sends $\mathcal{H}_e$ to the head node of $e$. Second, all intermediate nodes operate in the order defined by the natural partial ordering on $E'$ as follows: After an intermediate node $v$ has received Hilbert spaces from all edges whose head node is $v$, for all edges $e \in E'$ whose tail node is $v$, node $v$ operates the unitary operator $U_e$ on $\mathcal{H}_{In(e)} \otimes \mathcal{H}_e$ and sends $\mathcal{H}_e$ to the head node of $e$. Finally, after all edges have transmitted Hilbert spaces, for all $i$ satisfying $1 \le i \le m$, target node $t_i$ operates the unitary operator $V_i$ on $\mathcal{H}_{In(t_i)} \otimes \mathcal{V}_i$.
**Step 3: Measurement on Fourier basis**
The source node $s$ measures the Hilbert space $\mathcal{H}_s$ in the Fourier basis and sends the measurement outcome to all the terminal nodes $t_i$. For all edges $e \in E'$, the head node of $e$ measures the Hilbert space $\mathcal{H}_e$ in the Fourier basis and sends the measurement outcome to all terminal nodes $t_i$.
**Step 4: Recovery**
All terminal nodes $t_i$ operate $Z(c_1) \otimes \cdots \otimes Z(c_r)$ on $\mathcal{V}_i$. Here, $\{c_k\}_{k=1}^r$ is a set of natural numbers that can be determined from the measurement outcomes received in step 3, the classical linear multicast network code $\{f_e\}_{e \in E'}$ and the decoding operators $\{g_i\}_{i=1}^N$ [44].

---

In Step 3 of Protocol 1, the Fourier basis of $\{|\tilde{z}\rangle\}_{z \in \mathbb{F}_q} \subset \mathcal{H}_e$ of the computational basis $\{|x\rangle\}_{x \in \mathbb{F}_q} \subset \mathcal{H}_e$ is defined as

$$|\tilde{z}\rangle := \sum_{x \in \mathbb{F}_q} \omega^{\mathrm{Tr}\, xz} |x\rangle,$$

where $\omega := \exp(-2\pi i / p)$. Here, $\mathrm{Tr}\, z$ represents the element $\mathrm{Tr}\, M_z \in \mathbb{F}_p$, where $M_z$ is the matrix representation of the multiplication map $x \mapsto zx$. Here, we note that the finite field $\mathbb{F}_q$ can be identified with the vector space $\mathbb{F}_p^t$, where $t$ is the degree of the algebraic extension of $\mathbb{F}_q$. For further details, see [101], Section 8.1.2 . We also define the generalized Pauli operators $Z(t)$ as $Z(t) := \sum_{x \in \mathbb{F}_q} \omega^{\mathrm{Tr}\, xt} |x\rangle\langle x|$.

## 3. Results

In this section, we give all results of this paper. We give results for $1 \to 2$ and $1 \to 3$ asymmetric UQCs in the Sections 3.1 and 3.2, respectively.

### 3.1. $1 \to 2$ Asymmetric UQC Multicast Protocol

In this subsection, we present a new protocol that multicasts optimal asymmetric UQCs from the source node $s$ to two target nodes $t_1$ and $t_2$ on a quantum network. We present the protocol in Section 3.1.1 and prove that the it creates optimal asymmetric UQCs in Section 3.1.2.

### 3.1.1. $1 \rightarrow 2$ Quantum Multicast Protocol

In this sub-subsection, we present the protocol for multicasting $1 \rightarrow 2$ asymmetric optimal UQCs of an input quantum state from the source node $s$ to two target nodes $t_1$ and $t_2$.

As we have explained in Section 2.3, the problem settings for the protocol of Kobayashi et al. and our protocol are essentially the same, and only their purposes are different. Here, we summarize the problem setting of our quantum multicast network coding: A quantum network is described by an *undirected* graph $G = (V, E)$. There exist a source node $s \in V$ and $N$ target nodes $t_1, \ldots, t_N \in V$ on the network. In this subsection, since we consider multicasting $1 \rightarrow 2$ asymmetric UQCs, we set $N = 2$. In a single session, any quantum channel $(u, v) \in E$ can send a $q$-dimensional quantum system $\mathcal{H}_e$ just once, either from $u$ to $v$ or from $v$ to $u$, where $q$ is assumed to be a prime power. Furthermore, any quantum operations can be implemented on any node $v \in V$, and measurement outcomes can be freely sent among nodes. At the beginning of a session, a single copy of input state $|\psi\rangle$ is given on the source node $s$.

Under these problem settings, the purpose of our protocol is to construct optimal asymmetric universal clones given by Equation (7) between target nodes $t_1$ and $t_2$ for a given $d$-dimensional unknown input state $|\psi\rangle = \sum_{j=0}^{d} \alpha_j |j\rangle \in \mathcal{H}_s$ on a source node, where $\mathcal{H}_s$ is a $d$-dimensional input space. In other words, we consider multicast communication protocols that are optimal in terms of fidelity. Thus, our research problem is to construct a protocol with a smaller communication cost under the above condition. We assume $d = q^r$. In other words, we consider multicast quantum network coding with source rate $r$. Here, note that since we assumed $q$ is a prime power, $d$ is also a prime power.

For this purpose, we use two additional assumptions: The first assumption is the same assumption that Kobayashi et al. used. That is, we assume that there exists a solvable classical linear multicast network code $\{f_e\}_{e \in E'}$ with source rate $r$ on an acyclic directed graph $G' = (V, E')$ over a finite field $\mathbb{F}_q$, where $G$ is an undirected underlying graph of $G'$. Hence, we can use the quantum multicast network coding protocol of Kobayashi et al. with source rate $r$ on this quantum network $G$. We further assume that at most 2 ebits of entanglement resource are shared between target node $t_1$ and $t_2$. Hence, the amount of this entanglement resource is constant with respect to the dimension $d$ of the input state and is negligible for large $d$ in comparison to $d$.

Before we present the protocol, we define the unitary operators used in it. Pauli operators $X_d$ and $Z_d$ are defined as

$$X_d := \sum_{k=0}^{d-1} |k \oplus 1\rangle \langle k|, \quad Z_d := \sum_{k=0}^{d-1} \omega^k |k\rangle \langle k|, \tag{20}$$

where $\omega := e^{\frac{2\pi i}{d}}$. In the following part of the paper, unitary operators defined on $\mathbb{C}^d \otimes \mathbb{C}^d$ and $\mathbb{C}^d \otimes \mathbb{C}^d \otimes \mathbb{C}^d$ are called bipartite and tripartite unitary operators, respectively. $Y^{(r)}$ is defined as a bipartite unitary operator satisfying

$$Y^{(r)}(\cos \eta |jr\rangle + \sin \eta |rj\rangle) = |jr\rangle, \quad Y^{(r)}|rr\rangle = |rr\rangle,$$
$$Y^{(r)}(\sin \eta |jr\rangle - \cos \eta |rj\rangle) = |rj\rangle \tag{21}$$

for all $j \in \{0, \ldots, d-1\}$ satisfying $j \neq r$, and

$$Y^{(r)}|ij\rangle = |ij\rangle \tag{22}$$

for all $i, j \in \{0, \ldots, d-1\}$ satisfying $i, j \neq r$, where $\eta$ is defined by

$$\cos \eta = \frac{a}{\sqrt{1 - \frac{2ab}{d}}} \quad \text{and} \quad \sin \eta = \frac{b}{\sqrt{1 - \frac{2ab}{d}}}. \tag{23}$$

The bipartite unitary operator $V^{(r)}$ is defined by

$$V^{(r)} := \sum_{j \neq r} |j\rangle\langle j| \otimes U_{r,j} + |r\rangle\langle r| \otimes I \tag{24}$$

where the unitary operator $U_{r,j}$ is defined by

$$U_{r,j} = I - |j\rangle\langle j| - |r-1\rangle\langle r-1| + |j\rangle\langle r-1| + |r-1\rangle\langle j|.$$

The bipartite unitary operator $\Delta^{(r)}$ is defined by

$$\Delta^{(r)} := |r\rangle\langle r| \otimes X_d^{-(r-1)} + \sum_{j \neq r} |j\rangle\langle j| \otimes I. \tag{25}$$

The unitary operator $\Gamma^{(r)}$ on $\mathbb{C}^d \otimes \mathbb{C}^d \otimes \mathbb{C}^2$ is defined by

$$\Gamma^{(r)} := |r\rangle\langle r| \otimes \text{swap} + \sum_{j \neq r} |j\rangle\langle j| \otimes I, \tag{26}$$

where swap is a unitary operator on $\mathbb{C}^d \otimes \mathbb{C}^2$ defined by

$$\text{swap} := \sum_{i=2}^{d-1} \sum_{j=0,1} |ij\rangle\langle ij| + \sum_{i,j=0,1} |ij\rangle\langle ji|.$$

The unitary operator $\Theta$ on $\mathbb{C}^2 \otimes \mathbb{C}^2$ is defined by

$$\begin{aligned}
\Theta|jj\rangle &= |jj\rangle \quad (j = 0, 1) \\
\Theta(\cos\eta|01\rangle + \sin\eta|10\rangle) &= |10\rangle \\
\Theta(\sin\eta|01\rangle - \cos\eta|10\rangle) &= |01\rangle
\end{aligned} \tag{27}$$

The bipartite unitary operator $\Lambda^{(r)}$ is defined by

$$\Lambda^{(r)} := \sum_{j \neq r} |j\rangle\langle j| \otimes I + |r\rangle\langle r| \otimes X_d^r \tag{28}$$

Before starting the protocol, we prepare three $d$-dimensional systems $A$, $B$, and $M$ at the source node $s$, $d$-dimensional systems $C$, $E$, and 2-dimensional systems $G$, $T_1$ at the target node $t_1$. Similarly, we prepare $d$-dimensional systems $D$, $F$, and 2-dimensional systems $H$, $T_2$ at $t_2$. The entanglement resource $\cos\eta|0\rangle_E|1\rangle_F + \sin\eta|1\rangle_E|0\rangle_F$ is shared between $E$ and $F$, and the Bell state $\frac{1}{\sqrt{2}}(|00\rangle_{T_1 T_2} + |11\rangle_{T_1 T_2})$ is shared between $T_1$ and $T_2$. Thus, the amount of entanglement resources is at most 2 ebits.

The protocol for $1 \to 2$ is shown as Protocol 2. In Protocol 2, first, asymmetric UQCs of an $q^r$-dimensional input state $|\psi\rangle$ is created on the source node $s$ at Step 1. Then, by measuring an ancilla, the whole state is splitted into classical information (measurement result), which is sent to target nodes, and a bipartite quantum state at Step 2. The bipartite quantum state is compressed into a $q^r$-dimensional state at Step 3. The state is distributed to the target nodes $t_1$ and $t_2$ by the protocol of Kobayashi et al. at Step 4. From the GHZ-type state received in the previous step and the classical information received at Step 2, target nodes $t_1$ and $t_2$ reconstruct asymmetric UQCs of $|\psi\rangle$ by LOCC and preshared small entanglement resource; this process is completed from Step 5 to Step 12. As a result of the protocol, $1 \to 2$ asymmetric UQCs given by Equation (7) are created in systems $EF$, where $E$ and $F$ are on the target nodes $t_1$ and $t_2$, respectively. Note that as we explained in the previous subsection, optimal asymmetric UQCs depend on the parameters $a$ and $b$ in Equation (4).

We can set these parameters in Step 2 of the protocol, when we apply $U_{1 \to 2}^{(a,b)}$. As we already explained,

---

**Protocol 2** $1 \to 2$ quantum multicast network coding protocol

---

**Step 1:** At the beginning, the source node $s$ has an unknown input quantum state $|\psi\rangle_A$ on system $A$ and makes $1 \to 2$ asymmetric universal clones by applying an isometry $U_{1 \to 2}^{(a,b)}$ defined by Equation (4) from the system $A$ to the system $ABM$.

**Step 2:** The source node $s$ measures system $M$ in the computational basis and sends the measurement outcome $r$ to the two target nodes $t_1$ and $t_2$.

**Step 3:** The source node $s$ applies the unitary operator $Y_{AB}$ defined by Equations (21) and (22) to the systems $AB$, then discards the system $B$.

**Step 4:** The state on system $A$ is multicast to the target nodes $t_1$ and $t_2$ over the quantum network $G$ using the protocol of Kobayashi et al. The target nodes $t_1$ and $t_2$ put the output GHZ-type state of the protocol of Kobayashi et al. on system $CD$.

**Step 5:** The target nodes $t_1$ and $t_2$ apply $X_d^{r-1} \otimes X_d^{r-1}$ to system $EF$ using the measurement outcome $r$ sent from the source node $s$.

**Step 6:** The target node $t_1$ applies $V_{C,E}^{(r)}$ defined by Equation (24) to system $CE$, and the target node $t_2$ applies $V_{D,F}^{(r)}$ to system $DF$. Then, the target node $t_1$ applies $\Delta_{C,E}^{(r)}$ defined by Equation (25) to system $CE$, and the target node $t_2$ applies $\Delta_{D,F}^{(r)}$ to system $DF$.

**Step 7:** The target node $t_1$ initializes $G$ in $|0\rangle$ and applies $\Gamma_{C,E,G}^{(r)}$ defined by Equation (26) on system $CEG$. The target node $t_2$ initializes $H$ in $|0\rangle$ and applies $\Gamma_{C,E,G}^{(r)}$ to system $DFH$.

**Step 8:** The target node $t_2$ sends the state on system $H$ to system $T_1$ at the target node $t_1$ using the Bell state on system $T_1 T_2$ by the quantum teleportation.

**Step 9:** The target node $t_1$ applies $\Theta_{GT_1}$ defined by Equation (27) to systems $G$ and $T_1$ and discards $T_1$.

**Step 10:** The target node $t_1$ measures system $G$ in

$$\left\{ |\tilde{0}\rangle = \frac{|0\rangle + |1\rangle}{\sqrt{2}}, |\tilde{1}\rangle = \frac{|0\rangle - |1\rangle}{\sqrt{2}} \right\}$$

and derives the measurement outcome $k$. Then, $t_1$ performs $(I - 2|r\rangle\langle r|)^k$ on system $C$.

**Step 11:** The target node $t_1$ applies $\Lambda_{CE}^{(r)}$ defined by Equation (28) on the system $CE$, and the target node $t_2$ applies $\Lambda_{DF}^{(r)}$ the system $DF$.

**Step 12:** The target nodes $t_1$ and $t_2$ measure system $C$ and $D$ in the Fourier basis

$$\left\{ \sum_{x=0}^{d-1} \frac{\omega^{px}}{\sqrt{d}} |x\rangle \right\}_{p \in Z_d},$$

respectively, and derive the measurement outcomes $p_1$ and $p_2$, respectively. Then, $t_1$ applies $Z_d^{p_1+p_2}$ to system $E$, and $t_2$ applies $Z_d^{p_1+p_2}$ to system $F$.

---

3.1.2. Proof of $1 \to 2$ Quantum Multicast Protocol

In this sub-subsection, we present the proof that Protocol 2 creates $1 \to 2$ asymmetric UQCs given by Equation (7) in system $EF$ shared by target nodes $t_1$ and $t_2$.

**Proof.** As we explained in the previous subsection, an input state at the source node $s$ can be written as

$$|\psi\rangle = \sum_{j=0}^{d-1} \alpha_j |j\rangle \in \mathcal{H}_s.$$

Then, from Equation (4), the state on system $ABM$ after step 1 can be written as:

$$a|\psi\rangle_A |\Phi^+\rangle_{BM} + b|\psi\rangle_B |\Phi^+\rangle_{AM} \tag{29}$$

The unnormalized state $\left|\Psi_2^{(r)}\right\rangle_{AB}$ on system $AB$ after deriving measurement outcome $r$ in Step 2 can be written as:

$$\left|\Psi_2^{(r)}\right\rangle_{AB} := \beta_r|rr\rangle_{AB} + \sum_{j \neq r} \beta_j(\cos\eta|jr\rangle_{AB} + \sin\eta|rj\rangle_{AB}), \tag{30}$$

where $\eta$ is defined by Equation (23), and $\{\beta_j\}_{j=0}^{d-1}$ is defined by

$$\beta_r = \frac{\alpha_r}{\sqrt{d}}(a+b)$$

$$\beta_j = \frac{\alpha_j}{\sqrt{d}}\sqrt{1 - \frac{2ab}{d}} \quad (\forall j \neq r). \tag{31}$$

here, $\left\|\left|\Psi_2^{(r)}\right\rangle_{AB}\right\|^2 = \sum_j |\beta_j|^2$ is a probability in which outcome $r$ is derived in Step 2. Since measuring system $M$ without seeing the outcome is mathematically equivalent to tracing out system $M$, $\{\left|\Psi_2^{(r)}\right\rangle\}_{r=0}^{d-1}$ satisfies

$$\varepsilon_{1\to2}(|\psi\rangle\langle\psi|) = \sum_{r=0}^{d-1} \left|\Psi_2^{(r)}\right\rangle\left\langle\Psi_2^{(r)}\right|, \tag{32}$$

where $\varepsilon_{1\to2}$ is a $1 \to 2$ optimal asymmetric UQCM defined by Equation (7). Hence, the purpose of the remaining part of the protocol is to transfer $\left|\Psi_2^{(r)}\right\rangle$ to the target nodes. However, in our problem settings, the throughput of the quantum network is too small to send $\left|\Psi_2^{(r)}\right\rangle$ directly to the target nodes. Hence, first, we compress the state on the $d$-dimensional system in Step 3. Then, the unnormalized state of system $AB$ after Step 3 can be written as

$$|\Psi_3\rangle_A = \sum_{j=0}^{d-1} \beta_j|j\rangle_A. \tag{33}$$

In Step 4, the protocol of Kobayashi et al. successfully works based on the assumption for the existence of a classical linear multicast network code. Since the (unnormalized) input state for the protocol of Kobayashi et al. is $|\Psi_3\rangle_A$, the unnormalized state on the system $C$ at the target node $t_1$ and on system $D$ at the target node $t_2$ can be written as $\sum_{j=0}^{d-1} \beta_j|j\rangle_C|j\rangle_D$. The purpose of the remaining part of the protocol is to reconstruct $\left|\Psi_2^{(r)}\right\rangle$ from this state.

Since system $EF$ is initially on $\cos\eta|0\rangle_E|1\rangle_E + \sin\eta|1\rangle_E|0\rangle_F$, the unnormalized state on system $CDEF$ can be written as

$$\sum_{j=0}^{d-1} \beta_j|jj\rangle_{CD} \otimes (\cos\eta|0\rangle_E|1\rangle_F + \sin\eta|1\rangle_E|0\rangle_F) \tag{34}$$

Then, the unnormalized state on $CDEF$ after Step 5 can be written as

$$\sum_{j=0}^{d-1} \beta_j|jj\rangle_{CD} \otimes (\cos\eta|r-1\rangle_E|r\rangle_F + \sin\eta|r\rangle_E|r-1\rangle_F). \tag{35}$$

The unnormalized state on $CDEF$ after Step 6 is

$$\sum_{j \neq r} \beta_j|jj\rangle_{CD} \otimes (\cos\eta|j\rangle_E|r\rangle_F + \sin\eta|r\rangle_E|j\rangle_F)$$

$$+\beta_r|rr\rangle_{CD} \otimes (\cos\eta|0\rangle_E|1\rangle_F + \sin\eta|1\rangle_E|0\rangle_F). \tag{36}$$

Then, the unnormalized state on $CDEFGH$ after Step 7 can be written as

$$\sum_{j\neq r}\beta_j|jj\rangle_{CD}\otimes(\cos\eta|j\rangle_E|r\rangle_F+\sin\eta|r\rangle_E|j\rangle_F)\otimes|00\rangle_{GH}$$

$$+\beta_r|rr\rangle_{CD}\otimes|00\rangle_{EF}\otimes(\cos\eta|0\rangle_G|1\rangle_H+\sin\eta|1\rangle_G|0\rangle_H). \tag{37}$$

Next, in Step 8, the state on the system $H$ is transferred to system $T_1$ by quantum teleportation [102]. Thus, the unnormalized state on $CDEFG$ after Step 9 can be written as

$$\sum_{j\neq r}\beta_j|jj\rangle_{CD}\otimes(\cos\eta|j\rangle_E|r\rangle_F+\sin\eta|r\rangle_E|j\rangle_F)\otimes|0\rangle_G$$

$$+\beta_r|rr\rangle_{CD}\otimes|00\rangle_{EF}\otimes|1\rangle_G. \tag{38}$$

Since system $G$ is effectively removed in Step 10, the unnormalized state on system $CDEF$ after Step 10 can be written as

$$\sum_{j\neq r}\beta_j|jj\rangle_{CD}\otimes(\cos\eta|j\rangle_E|r\rangle_F+\sin\eta|r\rangle_E|j\rangle_F)$$

$$+\beta_r|rr\rangle_{CD}\otimes|00\rangle_{EF}. \tag{39}$$

Then, the unnormalized state on $CDEF$ after Step 11 can be written as

$$\sum_{j\neq r}\beta_j|jj\rangle_{CD}\otimes(\cos\eta|j\rangle_E|r\rangle_F+\sin\eta|r\rangle_E|j\rangle_F)$$

$$+\beta_r|rr\rangle_{CD}\otimes|rr\rangle_{EF}. \tag{40}$$

In Step 12, after system $CD$ is measured in the Fourier basis $\{d^{-1/2}\cdot\sum_{x=0}^{d-1}\omega^{px}|x\rangle\}_{p\in Z_d}$ and is discarded, the unnormalized state on $EF$ for the measurement outcomes $p_1$ and $p_2$ can be written as

$$\sum_{j\neq r}\beta_j\omega^{-j(p_1+p_2)}(\cos\eta|j\rangle_E|r\rangle_F+\sin\eta|r\rangle_E|j\rangle_F)$$

$$+\beta_r\omega^{-r(p_1+p_2)}|rr\rangle_{EF} \tag{41}$$

Hence, after applying $Z^{p_1+p_2}\otimes Z^{p_1+p_2}$ on system $EF$, the unnormalized state on $EF$ becomes

$$\omega^{r(p_1+p_2)}\cdot\sum_{j=0,j\neq r}^{d-1}\beta_j(\cos\eta|j\rangle_E|r\rangle_F+\sin\eta|r\rangle_E|j\rangle_F)+\beta_r|rr\rangle_{EF}. \tag{42}$$

This unnormalized state coincides with $\left|\Psi_2^{(r)}\right\rangle$ defined by Equation (42) except a global phase. Since Equation (42) is the unnormalized state corresponding to the outcome $r$ in Step 2, the final state of this protocol can be written as $\sum_r\left|\Psi_2^{(r)}\right\rangle\left\langle\Psi_2^{(r)}\right|$. Hence, by Equation (32), the final states of protocol 2 on the target nodes $t_1$ and $t_2$ are $1\to2$ optimal asymmetric UQCs of the input state $|\psi\rangle$.  □

### 3.2. $1\to3$ *Optimal Asymmetric Quantum Universal Clones Multicast Protocol*

In this subsection, we present a protocol that multicasts optimal asymmetric UQCs from the source node $s$ to three target nodes $t_1$, $t_2$, and $t_3$ on a quantum network. We present the protocol in Sections 3.2.1 and 3.2.2, we prove that creates optimal asymmetric UQCs.

### 3.2.1. $1\to3$ Quantum Multicast Protocol

In this sub-subsection, we present a protocol that multicasts $1\to3$ optimal asymmetric UQCs of an unknown input quantum state from the source node $s$ to two target nodes $t_1$, $t_2$, and $t_3$.

The problem setting for the $1 \to 3$ quantum multicast protocol is almost the same as that of the $1 \to 2$ protocol given in the last subsection. Hence, we consider only the difference between these two problem settings. First, the number of target nodes is different. That is, in this subsection, a quantum network $G$ has three target nodes $t_1, t_2$, and $t_3$. The purpose of the protocol is to construct $1 \to 3$ optimal asymmetric universal clones given by Equation (12) among target nodes $t_1, t_2$, and $t_3$ for a unknown $d$-dimensional input state $|\psi\rangle = \sum_{j=0}^{d} \alpha_j |j\rangle \in \mathcal{H}_s$ on a source node, where $\mathcal{H}_s$ is a $d$-dimensional input space. We again assume $d = q^r$. In other words, we consider a mulcast quantum network code with source rate $r$. The assumption for the existence of a classical linear multicast network code is also similar. That is, a classical linear multicast network code is a code on $\mathbb{F}_q$ used to multicast from the node $s$ to the nodes $t_1, t_2, t_3$ on $G'$ with source rate $r$. The amount of entanglement shared among the target nodes is also different. In $1 \to 3$ case, we assume that at most $2 + 4 \log_2 3$ ebits are shared among the target nodes $t_1, t_2$, and $t_3$. Hence, the amount of this entanglement resource is constant with respect to the dimension $d$ of the input state.

Before we present the protocol, we define the unitary operators used in the protocol. In the following definitions, a subscript of an unitary operator denotes the step of the protocol where it is used. Thus, for example, we do not define $U_2$ since we will not use any unitary operation at Step 2 of the protocol. $U_3^{(r,s)}$ is a tripartite unitary operator satisfying the following conditions:

$$U_3^{(r,s)} \cdot \frac{\alpha|jrs\rangle + \beta|rjs\rangle + \gamma|rsj\rangle + \alpha|jsr\rangle + \beta|sjr\rangle + \gamma|srj\rangle}{\sqrt{2\alpha^2 + 2\beta^2 + 2\gamma^2}} = |j00\rangle, \quad (\forall j \neq r, s)$$

$$U_3^{(r,s)} \cdot \frac{(\alpha + \beta)|rrs\rangle + (\beta + \gamma)|srr\rangle + (\gamma + \alpha)|rsr\rangle}{\sqrt{(\alpha + \beta)^2 + (\beta + \gamma)^2 + (\gamma + \alpha)^2}} = |r00\rangle, \quad (43)$$

$$U_3^{(r,s)} \cdot \frac{(\alpha + \beta)|ssr\rangle + (\beta + \gamma)|rss\rangle + (\gamma + \alpha)|srs\rangle}{\sqrt{(\alpha + \beta)^2 + (\beta + \gamma)^2 + (\gamma + \alpha)^2}} = |s00\rangle.$$

$U_5^{(r,s)}$ is a bipartite unitary operator defined by

$$U_5^{(r,s)} := |r\rangle\langle r| \otimes I + |s\rangle\langle s| \otimes I + \sum_{j \neq r,s}^{d-1} |j\rangle\langle j| \otimes \left( \sum_{x=0}^{d-1} |\pi_{jrs}(x)\rangle\langle x| \right), \quad (44)$$

where $\pi_{jrs}$ is a permutation satisfying the following conditions:

$$\pi_{jrs}(0) = j, \quad \pi_{jrs}(1) = r, \quad \pi_{jrs}(2) = s \quad (45)$$

$U_6^{(r,s)}$ is a tripartite unitary operator defined by

$$U_6^{(r,s)} := |r\rangle\langle r| \otimes swap + |s\rangle\langle s| \otimes swap + \sum_{j \neq r,s} |j\rangle\langle j| \otimes I \otimes I, \quad (46)$$

where $swap$ is a swap operator defined by

$$swap := \sum_{i,j=0}^{d-1} |ij\rangle\langle ji|. \quad (47)$$

$U_7^{(r,s)}$ is a bipartite unitary operator defined by

$$U_7^{(r,s)} := \sum_{i \neq r,s} |i\rangle\langle i| \otimes I + |r\rangle\langle r| \otimes \sum_{j=0}^{d-1} |\pi''_{rs}(j)\rangle\langle j| + |s\rangle\langle s| \otimes \sum_{k=0}^{d-1} |\pi''_{sr}(k)\rangle\langle k|, \quad (48)$$

where $\pi''_{xy}$ is a permutation satisfying $\pi''_{xy}(0) = x$ and $\pi''_{xy}(1) = y$. $U_8$ is a unitary operator on $\mathbb{C}^3 \otimes \mathbb{C}^3 \otimes \mathbb{C}^3$ satisfying

$$
\begin{aligned}
U_8(\alpha''_1|001\rangle + \beta''_1|100\rangle + \gamma''_1|010\rangle) &= |000\rangle \\
U_8(\alpha'_1(|012\rangle + |021\rangle) + \beta'_1(|102\rangle + |201\rangle) + \gamma'_1(|120\rangle + |210\rangle)) &= |100\rangle,
\end{aligned}
\tag{49}
$$

where $\alpha'_1, \beta'_1, \gamma'_1, \alpha''_1, \beta''_1$, and $\gamma''_1$ are defined by

$$
\begin{aligned}
\alpha'_1 &= \frac{\alpha}{\sqrt{2\alpha^2 + 2\beta^2 + 2\gamma^2}}, \quad \beta'_1 = \frac{\beta}{\sqrt{2\alpha^2 + 2\beta^2 + 2\gamma^2}}, \quad \gamma'_1 = \frac{\gamma}{\sqrt{2\alpha^2 + 2\beta^2 + 2\gamma^2}} \\
\alpha''_1 &= \frac{\alpha + \beta}{\sqrt{(\alpha + \beta)^2 + (\beta + \gamma)^2 + (\gamma + \alpha)^2}}, \quad \beta''_1 = \frac{\beta + \gamma}{\sqrt{(\alpha + \beta)^2 + (\beta + \gamma)^2 + (\gamma + \alpha)^2}}, \\
\gamma''_1 &= \frac{\gamma + \alpha}{\sqrt{(\alpha + \beta)^2 + (\beta + \gamma)^2 + (\gamma + \alpha)^2}}.
\end{aligned}
\tag{50}
$$

$U_9^{(r,s,k)}$ is a unitary operator on $\mathbb{C}^d$ defined by

$$
U_9^{(r,s,k)} := \sum_{j \neq r,s} |j\rangle\langle j| + (-1)^k |r\rangle\langle r| + (-1)^k |s\rangle\langle s|.
\tag{51}
$$

$U_2'^{(r)}$ is a tripartite unitary operator satisfying

$$
\begin{aligned}
U_2'^{(r)} \cdot \frac{\alpha|jrr\rangle + \beta|rjr\rangle + \gamma|rrj\rangle}{\sqrt{\alpha^2 + \beta^2 + \gamma^2}} &= |j00\rangle, \quad (\forall j \neq r) \\
U_2'^{(r)}|rrr\rangle &= |r00\rangle
\end{aligned}
\tag{52}
$$

$U_5'^{(r)}$ is a bipartite unitary operator defined by

$$
U_5'^{(r)} := |r\rangle\langle r| \otimes I + \sum_{j \neq r} |j\rangle\langle j| \otimes \left( \sum_{x=0}^{d-1} |\pi_{jr}(x)\rangle\langle x| \right),
\tag{53}
$$

where $\pi_{jr}$ is a permutation satisfying

$$
\pi_{jr}(1) = r, \quad \pi_{jr}(0) = j
\tag{54}
$$

$U_6'^{(r)}$ is a tripartite unitary operator defined by

$$
U_6'^{(r)} := |r\rangle\langle r| \otimes \mathrm{swap} + \sum_{j \neq r}^{d-1} |j\rangle\langle j| \otimes I \otimes I,
\tag{55}
$$

where swap is an operator defined by Equation (47). $U_7'^{(r)}$ is a tripartite unitary operator defined by

$$
U_7'^{(r)} := \sum_{i \neq r} |i\rangle\langle i| \otimes I + |r\rangle\langle r| \otimes X_d^r,
\tag{56}
$$

where $X_d$ is the Pauli $X$ operator defined by Equation (20). $U_8'$ is a unitary operator on $\mathbb{C}^2 \otimes \mathbb{C}^2 \otimes \mathbb{C}^2$ defined by

$$
\begin{aligned}
U_8'|000\rangle &= |000\rangle \\
U_8'(\alpha'_2|011\rangle + \beta'_2|101\rangle + \gamma'_2|110\rangle) &= |100\rangle
\end{aligned}
\tag{57}
$$

Finally, $U_9'^{(r,k)}$ is a unitary operator on $\mathbb{C}^d$ defined by

$$U_9'^{(r,k)} := \sum_{j \neq r} |j\rangle\langle j| + (-1)^k |r\rangle\langle r|. \tag{58}$$

We will also use in the protocol the projective measurement $\{P_k\}_{k=0}^2$ defined by the following equations:

$$P_0 := |\tilde{0}\rangle\langle\tilde{0}|, P_1 := |\tilde{1}\rangle\langle\tilde{1}|, P_2 := I - |\tilde{0}\rangle\langle\tilde{0}| - |\tilde{1}\rangle\langle\tilde{1}|, \tag{59}$$

where $|\tilde{0}\rangle := \frac{|0\rangle + |1\rangle}{\sqrt{2}}, |\tilde{1}\rangle := \frac{|0\rangle - |1\rangle}{\sqrt{2}}$.

At the beginning of the protocol, the source node $s$ has five $d$-dimensional systems $A$, $B$, $C$, $R$, and $S$. The target node $t_1$ has three $d$-dimensional systems $D$, $M_1$, and $N_1$. The target node $t_2$ has three $d$-dimensional systems $E$, $M_2$, and $N_2$. The target node $t_3$ has three $d$-dimensional systems $F$, $M_3$, and $N_3$. Furthermore, the target nodes $t_1$ and $t_2$ share $1 + 2\log_2 3$ ebits of entanglement, and the target nodes $t_1$ and $t_2$ share $1 + 2\log_2 3$ ebits of entanglement in the form of maximally entangled states. Hence, the amount of entanglement resources are $2 + 4\log_2 3$ ebits in total.

The beginning of the protocol for $1 \to 3$ is given in Protocol 3 . In Protocol 3 , first, asymmetric UQCs of an $q^r$-dimensional input state $|\psi\rangle$ is created on the source node $s$ at Step 1. Then, by measuring ancillary systems $R$ and $S$, the whole state is splitted into classical information (measurement results $r$ and $s$), which is sent to target nodes and a tripartite quantum state at Step 2. The continuation of the protocol branches depending on whether $r \neq s$ or $r = s$. The continuation for $r \neq s$ is given in Protocol 4 , and for $r = s$ is given in Protocol 5 . In both Protocols 4 and 5 , the tripartite quantum state is compressed into a $q^r$-dimensional state at Step 3. The state is distributed to the target nodes $t_1$, $t_2$, and $t_3$ by the protocol of Kobayashi et al. at Step 4. From the GHZ-type state received in the previous step and the classical information received at Step 2, target nodes $t_1$, $t_2$, and $t_3$ reconstruct asymmetric UQCs of $|\psi\rangle$ by LOCC and preshared small entanglement resource; this process is completed in the remaining part of the protocols. As the result of the protocol, $1 \to 3$ asymmetric UQCs given by Equation (12) are created system $M_1 M_2 M_3$, where $M_1$, $M_2$, and $M_3$ are on the target nodes $t_1$, $t_2$, and $t_3$, respectively. Note that as we explained in the previous subsection, asymmetric UQCs depends on the parameters $\alpha$, $\beta$, and $\gamma$ in Equation (10). We can set these parameters in step 1 of the protocol, when we apply $U_{1\to3}^{(\alpha,\beta,\gamma)}$.

---

**Protocol 3** $1 \to 3$ quantum multicast network coding protocol (beginning)

---

**Step 1:** At the beginning, the source node $s$ has an unknown input quantum state $|\psi\rangle_A$ on system $A$, and makes $1 \to 3$ asymmetric universal clones by applying an isometry $U_{1\to3}^{(\alpha,\beta,\gamma)}$ defined by Equation (10) from system $A$ to system $ABCRS$.

**Step 2:** The source node $s$ measures the systems $R$ and $S$ in the computational basis, where the measurement outcomes of $R$ and $S$ are called $r$ and $s$, respectively. The source node $s$ sends the measurement outcomes $r$ and $s$ to the target nodes $t_1$, $t_2$, and $t_3$. The following steps of the protocol depend on whether $r \neq s$ or $r = s$.

---

---

**Protocol 4** Continuation of Protocol 3 for $1 \to 3$ quantum multicast network coding (for $r \neq s$)

---

$[r \neq s]$

**Step 3:** The source node $s$ applies unitary operator $U_2^{(r,s)}$ defined by Equation (43) to system $ABC$, and then, discards systems $B$ and $C$.

**Step 4:** The state on system $A$ is multicast to the target nodes $t_1$, $t_2$, and $t_3$ over the quantum network $G$ using the protocol of Kobayashi et al. The target nodes $t_1$, $t_2$, and $t_3$ put the output of the protocol of Kobayashi et al. on system $DEF$. Then, using $2 \log_2 3$ ebits of entanglement, the targets nodes share the following state on system $M_1 M_2 M_3$:

$$\left( \alpha_1'(|012\rangle + |021\rangle) + \beta_1'(|102\rangle + |201\rangle) + \gamma_1'(|120\rangle + |210\rangle) \right)_{M_1,M_2,M_3},$$

where $\alpha_1'$, $\beta_1'$, and $\gamma_1'$ are defined by Equation (50). Furthermore, by using 2 ebits of entanglement, the target nodes share the following state on system $N_1 N_2 N_3$:

$$(\alpha_1''|001\rangle + \beta_1''|100\rangle + \gamma_1''|010\rangle)_{N_1,N_2,N_3}, \tag{60}$$

where $\alpha_1''$, $\beta_1''$ and $\gamma_1''$ are defined by Equation (50).

**Step 5:** The target nodes apply $U_{5,DM_1}^{(r,s)} \otimes U_{5,EM_2}^{(r,s)} \otimes U_{5,FM_3}^{(r,s)}$ to system $DM_1 EM_2 FM_3$, where $U_5^{(r,s)}$ is defined by Equation (44).

**Step 6:** The target nodes apply $U_{6,DM_1N_1}^{(r,s)} \otimes U_{6,EM_2N_2}^{(r,s)} \otimes U_{6,FM_3N_3}^{(r,s)}$ to system $DM_1N_1EM_2N_2FM_3N_3$, where $U_6^{(r,s)}$ is defined by Equation (46).

**Step 7:** The target nodes apply $U_{7,DM_1}^{(r,s)} \otimes U_{7,EM_2}^{(r,s)} \otimes U_{7,FM_3}^{(r,s)}$ to system $DM_1 EM_2 FM_3$, where $U_7^{(r,s)}$ is defined by Equation (48).

**Step 8:** Using $2\log_2 3$ ebits of entanglement resource, subspaces spanned by $\{|0\rangle, |1\rangle, |2\rangle\}$ of the systems $N_2$ and $N_3$ are sent from the target nodes $t_2$ and $t_3$ to the target node $t_1$, respectively. The target node $t_1$ applies $U_{8,N_1N_2N_3}$ to system $N_1N_2N_3$ and discards systems $N_2$ and $N_3$.

**Step 9:** The target node $t_1$ applies the projective measurement $\{P_k\}_{k=0}^2$ defined by Equation (59) on system $N_1$ in the basis and discards the quantum system $N_1$. Then, depending on the measurement outcome $k$, the target node $t_1$ applies $U_9^{(r,s,k)}$ defined by Equation (51) on system $D$.

**Step 10:** The target nodes $t_1$, $t_2$, and $t_3$ measure system $D$, $E$, and $F$ in the Fourier basis $\{d^{-1/2} \cdot \sum_{x=0}^{d-1} \omega^{px}|x\rangle\}_{p \in \mathbb{Z}_d}$, respectively. Then, they apply $Z^{(p_1+p_2+p_3)} \otimes Z^{(p_1+p_2+p_3)} \otimes Z^{(p_1+p_2+p_3)}$ to system $M_1 M_2 M_3$, where $p_1$, $p_2$, and $p_3$ are the measurement outcomes on the target nodes $t_1$, $t_2$, and $t_3$, respectively.

---

---

**Protocol 5** Continuation of Protocol 3 for $1 \to 3$ quantum multicast network coding (for $r = s$)

---

**[$r = s$]**

**Step 3:** The source node $s$ applies unitary operator $U_3^{\prime(r)}$ defined by Equation (52) to system $ABC$ and then discards the systems $B$ and $C$.

**Step 4:** The state on system $A$ is multicast to the target nodes $t_1$, $t_2$, and $t_3$ over the quantum network $G$ using the protocol of Kobayashi et al. The target nodes $t_1$, $t_2$, and $t_3$ put the output of the protocol of Kobayashi et al. on system $DEF$. Then, using 2 ebits of entanglement, the target nodes share the following state on system $M_1 M_2 M_3$:

$$\left( \alpha_2' |011\rangle + \beta_2' |101\rangle + \gamma_2' |110\rangle \right)_{M_1 M_2 M_3}, \tag{61}$$

where $\alpha_2' = \frac{2\alpha}{\sqrt{(2\alpha)^2 + (2\beta)^2 + (2\gamma)^2}}$, $\beta_2' = \frac{2\beta}{\sqrt{(2\alpha)^2 + (2\beta)^2 + (2\gamma)^2}}$ and $\gamma_2' = \frac{2\gamma}{\sqrt{(2\alpha)^2 + (2\beta)^2 + (2\gamma)^2}}$.

Furthermore, they initialize all the systems $N_1$, $N_2$, and $N_3$ in $|0\rangle$.

**Step 5:** The target nodes apply $U_{5,DM_1}^{\prime(r)} \otimes U_{5,EM_2}^{\prime(r)} \otimes U_{5,FM_3}^{\prime(r)}$ to system $DM_1 EM_2 FM_3$, where $U_5^{\prime(r)}$ is defined by Equation (53).

**Step 6:** The target nodes apply $U_{6,DM_1 N_1}^{\prime(r)} \otimes U_{6,EM_2 N_2}^{\prime(r)} \otimes U_{6,FM_3 N_3}^{\prime(r)}$ to system $DM_1 N_1 EM_2 N_2 FM_3 N_3$, where $U_6^{\prime(r)}$ is defined by Equation (55).

**Step 7:** The target nodes apply $U_{7,DM_1}^{\prime(r)} \otimes U_{7,EM_2}^{\prime(r)} \otimes U_{7,FM_3}^{\prime(r)}$ to system $DM_1 EM_2 FM_3$, where $U_7^{\prime(r)}$ is defined by Equation (56).

**Step 8:** By using 2 ebits of entanglement resource, subspaces spanned by $\{|0\rangle, |1\rangle\}$ of the systems $N_2$ and $N_3$ are sent from the target nodes $t_2$ and $t_3$ to the target node $t_1$, respectively. The target node $t_1$ applies $U_{8,N_1 N_2 N_3}'$ as defined by Equation (57) to system $N_1 N_2 N_3$ and discards system $N_2$ and $N_3$.

**Step 9:** The target node $t_1$ applies the projective measurement $\{P_k\}_{k=0}^2$ defined by Equation (59) on system $N_1$ in the basis and discards the quantum system $N_1$. Then, depending on the measurement outcome $k$, the target node $t_1$ applies $U_9^{(r,k)}$ defined by Equation (58) on the system $D$.

**Step 10:** The target nodes $t_1$, $t_2$, and $t_3$ measure system $D$, $E$, and $F$ in the Fourier basis $\{d^{-1/2} \cdot \sum_{x=0}^{d-1} \omega^{px} |x\rangle\}_{p \in \mathbb{Z}_d}$, respectively. Then, they apply $Z_d^{(p_1 + p_2 + p_3)} \otimes Z_d^{(p_1 + p_2 + p_3)} \otimes Z_d^{(p_1 + p_2 + p_3)}$ to system $M_1 M_2 M_3$, where $p_1$, $p_2$, and $p_3$ are the measurement outcomes on the target nodes $t_1$, $t_2$, and $t_3$, respectively.

---

### 3.2.2. Proof of $1 \to 3$ Quantum Multicast Protocol

In this sub-subsection, we prove that Protocols 3–5 create $1 \to 3$ asymmetric UQCs given by Equation (12) in system $M_1 M_2 M_3$.

**Proof.** Let the input state at the source node be $|\psi\rangle = \sum_{j=0}^{d-1} \delta_j |j\rangle$. Then, from Equation (10), the state on system $ABCRS$ after Step 1 can be written as

$$
\begin{aligned}
\sqrt{\frac{d}{2d+2}} [ & \alpha |\psi\rangle_A (|\Phi^+\rangle_{BR} |\Phi^+\rangle_{CS} + |\Phi^+\rangle_{BS} |\Phi^+\rangle_{CR}) \\
+ & \beta |\psi\rangle_B (|\Phi^+\rangle_{AR} |\Phi^+\rangle_{CS} + |\Phi^+\rangle_{AS} |\Phi^+\rangle_{CR}) \\
+ & \gamma |\psi\rangle_C (|\Phi^+\rangle_{AR} |\Phi^+\rangle_{BS} + |\Phi^+\rangle_{AS} |\Phi^+\rangle_{BR}) ]
\end{aligned} \tag{62}
$$

After Step 2, the protocol branches depending on whether $r \neq s$ or $r = s$, where $r$ and $s$ are the measurement outcomes of system $R$ and $S$, respectively.

The unnormalized state $\left|\Psi_2^{(r,s)}\right\rangle$ after Step 2 for $r \neq s$ can be written as

$$
\begin{aligned}
&\left|\Psi_2^{(r,s)}\right\rangle \\
&= \frac{1}{\sqrt{2d(d+1)}}\Big[\alpha(|\psi\rangle_A|r\rangle_B|s\rangle_C + |\psi\rangle_A|s\rangle_B|r\rangle_C) + \beta(|r\rangle_A|\psi\rangle_B|s\rangle_C + |s\rangle_A|\psi\rangle_B|r\rangle_C) \\
&\qquad\qquad + \gamma(|r\rangle_A|s\rangle_B|\psi\rangle_C + |s\rangle_A|r\rangle_B|\psi\rangle_C)\Big] \\
&= \frac{1}{\sqrt{2d(d+1)}}\Big[\delta_r\big((\alpha+\beta)|rrs\rangle + (\beta+\gamma)|srr\rangle + (\gamma+\alpha)|rsr\rangle\big)_{ABC} \\
&\qquad\qquad + \delta_s\big((\alpha+\beta)|ssr\rangle + (\beta+\gamma)|rss\rangle + (\gamma+\alpha)|srs\rangle\big)_{ABC} \\
&\qquad\qquad + \sum_{j\neq r,s}\delta_j\big(\alpha|jrs\rangle + \beta|rjs\rangle + \gamma|rsj\rangle + \alpha|jsr\rangle + \beta|sjr\rangle + \gamma|srj\rangle\big)_{ABC}\Big].
\end{aligned}
\tag{63}
$$

The unnormalized state $\left|\Psi_2^{(r,r)}\right\rangle$ after Step 2 for $r = s$ can be written

$$
\begin{aligned}
&\left|\Psi_2^{(r,r)}\right\rangle \\
&= \sqrt{\frac{2}{d(d+1)}}\big[\alpha|\psi\rangle_A|r\rangle_B|r\rangle_C + \beta|r\rangle_A|\psi\rangle_B|r\rangle_C + \gamma|r\rangle_A|r\rangle_B|\psi\rangle_C\big] \\
&= \sqrt{\frac{2}{d(d+1)}}\Big[\delta_r(\alpha+\beta+\gamma)|rrr\rangle + \sum_{j\neq r}\delta_j\big(\alpha|jrr\rangle + \beta|rjr\rangle + \gamma|rrj\rangle\big)\Big].
\end{aligned}
\tag{64}
$$

As for the $1 \to 2$ quantum multicast network coding protocol, $\left\{\left|\Psi_2^{(r,s)}\right\rangle\right\}_{r,s=0}^{d-1}$ satisfies

$$
\epsilon_{1\to3}^{\alpha,\beta,\gamma}(|\psi\rangle\langle\psi|) = \sum_{r,s=0}^{d-1}\left|\Psi_2^{(r,s)}\right\rangle\left\langle\Psi_2^{(r,s)}\right|,
\tag{65}
$$

where $\epsilon_{1\to3}^{\alpha,\beta,\gamma}$ is a $1 \to 3$ optimal asymmetric UQCM defined by Equation (12). Hence, the purpose of the remaining part of the protocol is to transfer $\left|\Psi_2^{(r,s)}\right\rangle$ to the target nodes.

First, we give the continuation of the proof for $r \neq s$ (Protocol 4). We compress the state on a $d$-dimensional system on step 3. The unnormalized state on system $A$ after Step 3 can be written as

$$
\sum_{j=0}^{d-1}\kappa_j|j\rangle,
\tag{66}
$$

where $\{\kappa_j\}_{j=0}^{d-1}$ is defined as

$$
\begin{aligned}
\kappa_j &= \sqrt{\frac{d}{2d+2}}\frac{\delta_j}{d}\sqrt{2\alpha^2 + 2\beta^2 + 2\gamma^2} \qquad (j \neq r,s), \\
\kappa_r &= \sqrt{\frac{d}{2d+2}}\frac{\delta_r}{d}\sqrt{(\alpha+\beta)^2 + (\beta+\gamma)^2 + (\gamma+\alpha)^2}, \\
\kappa_s &= \sqrt{\frac{d}{2d+2}}\frac{\delta_s}{d}\sqrt{(\alpha+\beta)^2 + (\beta+\gamma)^2 + (\gamma+\alpha)^2}.
\end{aligned}
\tag{67}
$$

In Step 4, the protocol of Kobayashi et al. successfully works based on the assumption for the existence of a classical linear multicast network code. The unnormalized state on systems $DEF$ shared by the target nodes $t_1$, $t_2$, and $t_3$ can be written as

$$\sum_{j=0}^{d-1} \kappa_j |j\rangle_D |j\rangle_E |j\rangle_F.$$

Hence, the unnormalized state after Step 4 can be written as

$$\sum_{j=0}^{d-1} \kappa_j |jjj\rangle_{DEF} \otimes \left( \alpha_1' |012\rangle + \beta_1' |102\rangle + \gamma_1' |120\rangle + \alpha_1' |021\rangle + \beta_1' |201\rangle + \gamma_1' |210\rangle \right)_{M_1 M_2 M_3}$$

$$\otimes \left( \alpha_1'' |001\rangle + \beta_1'' |100\rangle + \gamma_1'' |010\rangle \right)_{N_1 N_2 N_3} \tag{68}$$

The purpose of the remaining part of the protocol is to reconstruct $\left| \Psi_2^{(r,s)} \right\rangle$ from the above state. The unnormalized state after Step 5 can be written as

$$\Big( \sum_{\substack{j \neq r,s}}^{d-1} \kappa_j |jjj\rangle_{DEF} \otimes (\alpha_1' |jrs\rangle + \beta_1' |rjs\rangle + \gamma_1' |rsj\rangle + \alpha_1' |jsr\rangle + \beta_1' |sjr\rangle + \gamma_1' |srj\rangle)_{M_1 M_2 M_3}$$

$$+ (\kappa_r |rrr\rangle_{DEF} + \kappa_s |sss\rangle_{DEF}) \tag{69}$$

$$\otimes (\alpha_1' |012\rangle + \beta_1' |102\rangle + \gamma_1' |120\rangle + \alpha_1' |021\rangle + \beta_1' |201\rangle + \gamma_1' |210\rangle)_{M_1 M_2 M_3} \Big)$$

$$\otimes (\alpha_1'' |001\rangle + \beta_1'' |100\rangle + \gamma_1'' |010\rangle)_{N_1 N_2 N_3}$$

The unnormalized state after Step 6 can be written as

$$\sum_{\substack{j \neq r,s}}^{d-1} \kappa_j |jjj\rangle_{DEF} \otimes (\alpha_1' |jrs\rangle + \beta_1' |rjs\rangle + \gamma_1' |rsj\rangle + \alpha_1' |jsr\rangle + \beta_1' |sjr\rangle + \gamma_1' |srj\rangle)_{M_1 M_2 M_3}$$

$$\otimes (\alpha_1'' |001\rangle + \beta_1'' |100\rangle + \gamma_1'' |010\rangle)_{N_1 N_2 N_3} \tag{70}$$

$$+ (\kappa_r |rrr\rangle + \kappa_s |sss\rangle)_{DEF} \otimes (\alpha_1'' |001\rangle + \beta_1'' |100\rangle + \gamma_1'' |010\rangle)_{M_1 M_2 M_3}$$

$$\otimes (\alpha_1' |012\rangle + \beta_1' |102\rangle + \gamma_1' |120\rangle + \alpha_1' |021\rangle + \beta_1' |201\rangle + \gamma_1' |210\rangle)_{N_1 N_2 N_3}$$

Then, the unnormalized state after Step 7 can be written as

$$\sum_{\substack{j \neq r,s}}^{d-1} \kappa_j |jjj\rangle_{DEF} \otimes (\alpha_1' |jrs\rangle + \beta_1' |rjs\rangle + \gamma_1' |rsj\rangle + \alpha_1' |jsr\rangle + \beta_1' |sjr\rangle + \gamma_1' |srj\rangle)_{M_1 M_2 M_3}$$

$$\otimes (\alpha_1'' |001\rangle + \beta_1'' |100\rangle + \gamma_1'' |010\rangle)_{N_1 N_2 N_3}$$

$$+ \kappa_r |rrr\rangle_{DEF} \otimes (\alpha_1'' |rrs\rangle + \beta_1'' |srr\rangle + \gamma_1'' |rsr\rangle)_{M_1 M_2 M_3} \tag{71}$$

$$\otimes (\alpha_1' |012\rangle + \beta_1' |102\rangle + \gamma_1' |120\rangle + \alpha_1' |021\rangle + \beta_1' |201\rangle + \gamma_1' |210\rangle)_{N_1 N_2 N_3}$$

$$+ \kappa_s |sss\rangle_{DEF} \otimes (\alpha_1'' |ssr\rangle + \beta_1'' |rss\rangle + \gamma_1'' |srs\rangle)_{M_1 M_2 M_3}$$

$$\otimes (\alpha_1' |012\rangle + \beta_1' |102\rangle + \gamma_1' |120\rangle + \alpha_1' |021\rangle + \beta_1' |201\rangle + \gamma_1' |210\rangle)_{N_1 N_2 N_3}$$

The unnormalized state after Step 8 can be written as

$$\sum_{\substack{j \neq r,s}}^{d-1} \kappa_j |jjj\rangle_{DEF} \otimes (\alpha_1' |jrs\rangle + \beta_1' |rjs\rangle + \gamma_1' |rsj\rangle + \alpha_1' |jsr\rangle + \beta_1' |sjr\rangle + \gamma_1' |srj\rangle)_{M_1 M_2 M_3} \otimes |0\rangle_{N_1}$$

$$+ \kappa_r |rrr\rangle_{DEF} \otimes (\alpha_1'' |rrs\rangle + \beta_1'' |srr\rangle + \gamma_1'' |rsr\rangle)_{M_1 M_2 M_3} \otimes |1\rangle_{N_1} \tag{72}$$

$$+ \kappa_s |sss\rangle_{DEF} \otimes (\alpha_1'' |ssr\rangle + \beta_1'' |rss\rangle + \gamma_1'' |srs\rangle)_{M_1 M_2 M_3} \otimes |1\rangle_{N_1}$$

The unnormalized state after Step 9 can be written as

$$
\sum_{j \neq r,s}^{d-1} \kappa_j |jjj\rangle_{DEF} \otimes (\alpha_1'|jrs\rangle + \beta_1'|rjs\rangle + \gamma_1'|rsj\rangle + \alpha_1'|jsr\rangle + \beta_1'|sjr\rangle + \gamma_1'|srj\rangle)_{M_1 M_2 M_3}
$$

$$
+ \kappa_r |rrr\rangle_{DEF} \otimes (\alpha_1''|rrs\rangle + \beta_1''|srr\rangle + \gamma_1''|rsr\rangle)_{M_1 M_2 M_3}
$$

$$
+ \kappa_s |sss\rangle_{DEF} \otimes (\alpha_1''|ssr\rangle + \beta_1''|rss\rangle + \gamma_1''|srs\rangle)_{M_1 M_2 M_3}
\tag{73}
$$

The unnormalized state after Step 10 can be written as

$$
\omega^{(p_1+p_2+p_3)(r+s)} \Bigg\{ \sum_{j \neq r,s}^{d-1} \kappa_j (\alpha_1'|jrs\rangle + \beta_1'|rjs\rangle + \gamma_1'|rsj\rangle + \alpha_1'|jsr\rangle + \beta_1'|sjr\rangle + \gamma_1'|srj\rangle)_{M_1 M_2 M_3}
$$

$$
+ \kappa_r (\alpha_1''|rrs\rangle + \beta_1''|srr\rangle + \gamma_1''|rsr\rangle)_{M_1 M_2 M_3}
$$

$$
+ \kappa_s (\alpha_1''|ssr\rangle + \beta_1''|rss\rangle + \gamma_1''|srs\rangle)_{M_1 M_2 M_3} \Bigg\}
\tag{74}
$$

We can easily see that the above state is equivalent to $\left| \Psi_2^{(r,s)} \right\rangle$ as defined by Equation (63) except for a global phase. Hence, the proof is complete for $r \neq s$.

Next, we give the continuation of the proof for $r = s$ (Protocol 5) . Equation (64) guarantees that the unnormalized state on system $A$ after Step 3 can be written as

$$
\sum_{j=0}^{d-1} \kappa_j' |j\rangle,
\tag{75}
$$

where $\{\kappa_j\}_{j=0}^{d-1}$ is defined as

$$
\kappa_j' = \sqrt{\frac{2}{d(d+1)}} \delta_j \sqrt{\alpha^2 + \beta^2 + \gamma^2} \quad (j \neq r), \qquad \kappa_r' = \sqrt{\frac{2}{d(d+1)}} \delta_r (\alpha + \beta + \gamma).
\tag{76}
$$

In Step 4, the protocol of Kobayashi et al. successfully works, and the unnormalized state at the target nodes can be written as $\sum_{j=0}^{d-1} \kappa_j |j\rangle_D |j\rangle_E |j\rangle_F$. Hence, the unnormalized state after Step 4 can be written as

$$
\sum_{j=0}^{d-1} \kappa_j' |jjj\rangle_{DEF} \otimes (\alpha_2'|011\rangle + \beta_2'|101\rangle + \gamma_2'|110\rangle)_{M_1 M_2 M_3} \otimes |000\rangle_{N_1 N_2 N_3}
\tag{77}
$$

Then, the unnormalized state after Step 5 can be written as

$$
\sum_{j \neq r}^{d-1} \kappa_j' |jjj\rangle_{DEF} \otimes (\alpha_2'|jrr\rangle + \beta_2'|rjr\rangle + \gamma_2'|rrj\rangle)_{M_1 M_2 M_3} \otimes |000\rangle_{N_1 N_2 N_3}
$$

$$
+ \kappa_r' |rrr\rangle_{DEF} \otimes (\alpha_2'|011\rangle + \beta_2'|101\rangle + \gamma_2'|110\rangle)_{M_1 M_2 M_3} \otimes |000\rangle_{N_1 N_2 N_3}
\tag{78}
$$

The unnormalized state after Step 6 can be written as

$$
\sum_{j \neq r}^{d-1} \kappa_j' |jjj\rangle_{DEF} \otimes (\alpha_2'|jrr\rangle + \beta_2'|rjr\rangle + \gamma_2'|rrj\rangle)_{M_1 M_2 M_3} \otimes |000\rangle_{N_1 N_2 N_3}
$$

$$
+ \kappa_r' |rrr\rangle_{DEF} \otimes |000\rangle_{M_1 M_2 M_3} (\alpha_2'|011\rangle + \beta_2'|101\rangle + \gamma_2'|110\rangle)_{N_1 N_2 N_3}
\tag{79}
$$

The unnormalized state after Step 7 can be written as

$$\sum_{j\neq r}^{d-1}\kappa_j'|jjj\rangle_{DEF}\otimes(\alpha_2'|jrr\rangle+\beta_2'|rjr\rangle+\gamma_2'|rrj\rangle)_{M_1M_2M_3}\otimes|000\rangle_{N_1N_2N_3}$$
$$+\kappa_j'|rrr\rangle_{DEF}\otimes|rrr\rangle_{M_1M_2M_3}(\alpha_2'|011\rangle+\beta_2'|101\rangle+\gamma_2'|110\rangle)_{N_1N_2N_3}$$

$$(80)$$

Then, the unnormalized state after Step 8 can be written as

$$\sum_{j\neq r}^{d-1}\kappa_j'|jjj\rangle_{DEF}\otimes\left(\alpha_2'|jrr\rangle+\beta_2'|rjr\rangle+\gamma_2'|rrj\rangle\right)_{M_1M_2M_3}\otimes|0\rangle_{N_1}$$
$$+\kappa_r'|rrr\rangle_{DEF}\otimes|rrr\rangle_{M_1M_2M_3}|1\rangle_{N_1}$$

$$(81)$$

The unnormalized state after Step 9 can be written as

$$\sum_{j\neq r}^{d-1}\kappa_j'|jjj\rangle_{DEF}\otimes(\alpha_2'|jrr\rangle+\beta_2'|rjr\rangle+\gamma_2'|rrj\rangle)_{M_1M_2M_3}+\kappa_r'|rrr\rangle_{DEF}\otimes|rrr\rangle_{M_1M_2M_3}$$

$$(82)$$

Finally, the unnormalized state after Step 10 can be written as

$$\omega^{(p_1'+p_2'+p_3')2r}\left\{\sum_{j=0,j\neq r}^{d-1}\kappa_j'(\alpha_2'|jrr\rangle+\beta_2'|rjr\rangle+\gamma_2'|rrj\rangle)_{M_1M_2M_3}+\kappa_r'|rrr\rangle_{M_1M_2M_3}\right\}$$

$$(83)$$

We can easily see that the above state is equivalent to $\left|\Psi_2^{(r,r)}\right\rangle$ as defined by Equation (64) except for a global phase. Hence, the proof is complete for $r=s$. Thus, we have achieved multicasting of asymmetric optimal clones for the systems $M_1$, $M_2$, and $M_3$ to the three target nodes. $\square$

## 4. Discussion

In this section, we discuss the results derived in the previous section. We give a discussion about the comparison with a conventional scheme in the Section 4.1, the relationship with our protocol and quantum telecloning in Section 4.2, the possibility of the extension of the results in the subsection Section 4.3, and a summary of the results in the Section 4.4.

### 4.1. Comparison with a Conventional Schemes

In this subsection, we compare our protocol with a conventional protocol without network coding based on entanglement swapping (or quantum repeater [25,26,28,32]), where maximally entangled states are distilled between a source node and a target node; then, asymmetric optimal UQCs are sent by teleportation. For simplicity, we concentrate on the multicast butterfly network, which is given as an undirected underlying graph of the directed graph on the left-hand side of Figure 1. Suppose each channel on the quantum network can transmit a $q$-dimensional quantum system in a single session, where $q$ is a prime power, and the target nodes share 2 ebit of entanglement. Then, since the rate of the linear solvable network code given on the left-hand side of Figure 1 is 2, our protocol can multicast an asymmetric optimal UQC of a $q^2$-dimensional unknown input state from source node $s$ to target nodes $t_0$ and $t_1$.

When we do not use the network coding scheme, a conventional scheme may be given as follows: First, maximally entangled states are distilled between source node $s$ and target nodes $t_0$ and $t_1$. Then, quantum teleportation is implemented to send an asymmetric optimal UQC from source node $s$ to target nodes $t_0$ and $t_1$. Since there are only two channels connected to the source node, there is no way to share more than $2\log_2 q$ ebit between the source node and the other nodes of the networks [25,26,28].

Hence, the best strategy in the first step is that $\log_2 q$ ebit is shared between the source node share and target node $t_0$, and other $\log_2 q$ ebit is also shared between the source node share and target node $t_1$. Thus, the source node can only teleport an asymmetric optimal UQC of a $q$-dimensional unknown input state to the target nodes. Therefore, the rate of the conventional scheme is just a half of our scheme. We note that 2 ebit of entanglement shared between the target nodes is not used in this conventional scheme.

### 4.2. Relation to Quantum Telecloning

In this subsection, we discuss the relationship with our protocol and quantum telecloning [99,100,103–105]. We first give a short review of quantum telecloning. Quantum telecloning is a protocol to multicast an optimal UQC from a sender to multiple receivers by local operation and classical communication with help of a preshared entangled state. There are various different variations of telecloning protocols [99,100,103–105]. Among them, the one that is strongly related to our problem is Ghiu's protocol to multicast $1 \to 2$ asymmetric optimal UQCs [99]. In Ghiu's protocol, the state $|Y\rangle_{RABM}$ defined by

$$|Y\rangle_{RABM} := I_R \otimes U_{1\to2}^{(a,b)}|\Phi_d^+\rangle_{RA}, \tag{84}$$

is shared among a sender and the first and second receivers, where $U_{1\to2}^{(a,b)}$ is defined by Equation (4) and $|\Phi_d^+\rangle$ is a standard maximally entangled state. The sender, the first receiver, and the second receiver possess the system $R$, $A$, and $B$, respectively. The system $M$ can be possessed by anyone since we do not need to apply any operation on $M$. The sender further possesses an unknown state $|\psi\rangle$ on the additional system $S$. At the first step of Ghiu's protocol, the sender applies a generalized Bell measurement on $RS$, and sends the measurement outcome to the receivers. Then, the resulted state on $ABM$ is an asymmetric optimal UQCs given by Equation (4) with an error depending on the measurement outcomes. Ghiu proved that there exist local unitary operations on $ABM$ that corrects this error [99]. As a result of the local unitary operations, the asymmetric optimal UQCs are shared between the target nodes. Note that since $M$ is just an ancillary system, we do not necessarily correct the error.

If we consider a way to use Ghiu's protocol for our problem on the multicast butterfly network, the problem reduces to finding an efficient way to share $|Y\rangle_{RABM}$ among the source node and the two target nodes. A conventional strategy to share this state may be as follows: First, maximally entangled states are shared among the source node and target nodes by using quantum channels on the network. Second, $|Y\rangle_{RABM}$ is prepared on the source node. Finally, the system $A$ and $B$ are teleported to the target nodes by using the maximally entangled states. However, this protocol is almost the same as the conventional protocol presented in the previous subsection, which is the protocol just teleporting asymmetric optimal QCMs using shared maximally entangled states. Thus, the rate of this protocol is also half of the rate of our protocol using network coding.

We can easily see that the system $M$ does not need to be used on Ghiu's protocol, and his telecloning protocol works just using the state $\rho_{RAB}$ defined by

$$\rho_{RAB} := \text{Tr}_M |Y\rangle\langle Y|_{RABM}. \tag{85}$$

Our protocol can be used to share the state $\rho_{RAB}$ among the source node and the two target nodes as follows. Suppose that there is a quantum network satisfying the assumption used in our $1 \to 2$ protocol. That is, it has a source node and two target nodes, the target nodes share 2 ebit, and the corresponding classical network has a classical solvable linear network code with rate $r$. Suppose each quantum channel on the quantum network can transmit $q$-dimensional system in a single session. Then, first, the source node prepares $|\Phi_{q^r}^+\rangle$ on the system $RA$. Second, the source node applies our protocol using the system $A$ as an input system. Then, we can easily see that as a result of our protocol, $\rho_{RAB}$ defined by Equation (85) is shared among the source node and the two target nodes. Here, we

emphasize that the dimension of $R$ ( as well as $A$ and $B$) is $q^r$. That is, the rate of this protocol is again $r$. For example, in the case of the multicast butterfly network, the rate is 2, which is double the conventional scheme that is explained above. Hence, our protocol can be used as an efficient preparation for asymmetric telecloning over quantum networks.

### 4.3. Possibility of the Extension of Our Protocol for More than 3 Receivers

In this subsection, we consider a possible extension of our protocol for more than 3 receivers. As a result, we suggest that there might not exist a straightforward extension of the protocol for an arbitrary number of terminal nodes using existing asymmetric optimal UQC schemes. In other words, we estimate that if we use an existing scheme, in order to multicast a $d$-dimensional unknown state, target nodes need to share $O(\log d)$ ebit of entanglement.

At first, we note that there are not so many existing works about asymmetric optimal UQCs outputting more than 3 copies. That is, Ren et al. and Ćwikliński et al. studied $1 \to 4$ asymmetric optimal UQCs [82,83], and Key et al. studied $1 \to N$ asymmetric optimal UQCs [81,84]. Since only existing $1 \to N$ asymmetric optimal UQC scheme is one given by Key et al., we focus on applying their scheme for multicast communication on quantum networks in this subsection.

Suppose $\mathcal{H}_I$ and $\mathcal{H}_O$ are a $d$-dimensional input space and a $d^N$ dimensional output space, respectively, and $N$ satisfies $N \geq 3$. Then, the $1 \to N$ asymmetric optimal UQC of Key et al. [84] is given by the following isometry $U_{I \to O}$ from $\mathcal{H}_I$ to $\mathcal{H}_O$:

$$U_{I \to O} := \sum_{i,j=0}^{d-1} \beta_1 |ji \cdots ii\rangle \langle j| + \beta_2 |ij \cdots ii\rangle \langle j| + \cdots + \beta_N |ii \cdots ij\rangle \langle j|, \tag{86}$$

where $\{\beta_n\}_{n=1}^N$ is an eigenvector corresponding to the maximum eigenvalue of matrix $A$ defined by

$$A := \sum_{n,m=1}^{N} \alpha_n |n\rangle \langle m| + (d-1) \sum_{n=1}^{N} \alpha_n |n\rangle \langle n|, \tag{87}$$

and satisfies the following normalization condition:

$$\left( \sum_{n=1}^{N} \beta_n \right)^2 + (d-1) \sum_{n=1}^{N} \beta_n^2 = d. \tag{88}$$

In the above equation, non-negative real parameters $\{\alpha_n\}_{n=1}^N$ represent an asymmetry of the UQC and satisfy $\sum_{n=1}^N \alpha_n = 1$. Equation (86) leads that an output state $|\Psi_O\rangle$ of this UQC protocol corresponding to a given input state $|\psi_I\rangle := \sum_{i=0}^{d-1} a_i |i\rangle$ can be written as

$$|\Psi_O\rangle = \sum_{i,j=0}^{d-1} a_j (\beta_1 |ji \cdots i\rangle + \beta_2 |ij \cdots i\rangle + \cdots + \beta_N |ii \cdots j\rangle). \tag{89}$$

We should note that even if $N = 3$, $U_{I \to O}$ defined by Equation (86) does not coincide the cloning map given by Equation (10); that is, the UQC protocol of Key et al. is not a straightforward extension of the UQC protocol used in this paper.

Let us assume $d >> N$, that is, the dimension $d$ of the input space $\mathcal{H}_I$ is much larger than the number of clones $N$, and give a rough estimation of the communication cost that is necessary to multicast $|\Psi_O\rangle$ to $N$ distinct terminal nodes on quantum networks. Here, we also assume $\alpha_1 > \alpha_n$ for all $n \geq 2$ for simplicity. Under this condition, $A$ can be approximated the diagonal matrix as

$$A \approx (d-1) \sum_{n=1}^{N} \alpha_n |n\rangle \langle n|. \tag{90}$$

Then, the maximum eigenvector $\{\beta_n\}_{n=1}^N$ of $A$ satisfying Equation (88) can be approximately given by $\beta_1 = 1$ and $\beta_n = 0$ for all $n \geq 2$. Hence, for a given input state $|\psi_I\rangle \in \mathcal{H}_I$, the output state $|\Psi_O\rangle \in \mathcal{H}_O$ can be approximated as

$$|\Psi_O\rangle \approx |\psi_I\rangle \otimes |GHZ_{N-1}\rangle, \tag{91}$$

where $|GHZ_{N-1}\rangle$ is the $N-1$ partite GHZ state on $\left(\mathbb{C}^d\right)^{\otimes N-1}$ defined as

$$|GHZ_{N-1}\rangle := \frac{1}{\sqrt{d}} \sum_{i=0}^{d-1} |ii \cdots i\rangle.$$

Let us consider a quantum network described by an undirected graph $G$ with one source node $s$ and $N$ target nodes $t_1, \cdots t_N$, and assume that the protocol can multicast a GHZ-type state $\sum_{i=0}^{d-1} a_i |ii \cdots i\rangle$ from the source node to the $N$ target nodes for a given input state $|\psi_I\rangle = \sum_{i=0}^{d-1} a_i |i\rangle$. Now, we consider the similar type of multicast protocol used in this paper. Thus, the protocol consists of the following three steps:

**Step 1** A quantum operation possibly including measurements are applied to $|\psi_I\rangle$ on $s$, where the measurement result (if they exist) is sent to $t_1, \cdots t_N$.

**Step 2** The output of the quantum operation, which should be $d$-dimensional system, is multicasted to the target nodes $t_1, \cdots t_N$ by the protocol of Kobayashi et al.

**Step 3** The state $|\Psi_O\rangle$ defined by Equation (89) is constructed by LOCC with an additional entanglement resource on $t_1, \cdots t_N$.

Now, we give a rough discussion that strongly suggests that the necessary entanglement resource on the target node is $O(\log d)$ ebit: In order to make the state $|\Psi_O\rangle$ given by Equation (89) on the target nodes, the protocol of Kobayashi et al. may be necessary to multicast the whole $d$-dimensional output space of the quantum operation on $s$ in Step 2. In other words, any quantum state $\sum_{i=0}^{d-1} a_i'|i\rangle$ on $\mathbb{C}^d$ is necessary to multicast by the protocol of Kobayashi et al. from $s$ to $t_1, \cdots t_N$, where $a_i'$ is not equal to $a_i$ in general. Then, when $a_0' = 1$ and $a_i' = 0$ for all $i \geq 1$, the output state of the protocol of Kobayashi et al.is a product state $|00 \cdots 0\rangle$. On the other hand, Equation (91) guarantees that when $d >> N$, the entanglement of the target state $|\Psi_O\rangle$ is $O(\log d)$ ebit for any bipartition on $\{1, \cdots, N\}$ except $1|2, \cdots N$. Therefore, we need an additional entanglement resource with an amount of at least $O(\log d)$ ebit for any bipartite except $1|2, \cdots N$.

This conclusion is completely different from the results in Sections 3.1 and 3.2, where $1 \to 2$ and $1 \to 3$ asymmetric optimal UQCs are constructed with one use of the protocol of Kobayashi et al. and an entanglement resource, *which is a constant with respect to the dimension d.* Hence, we conclude that by using existing asymmetric optimal UQC protocol, there might not exist a straightforward extension of our protocol for an arbitrary large number of terminal nodes.

### 4.4. Summary

In this paper, we considered quantum multicast network coding as the multicasting of optimal UQCs over a quantum network. By extending the results of Owari et al. [48–50] for a multicast of symmetric optimal UQCs, we developed a protocol to multicast asymmetric optimal UQCs over a quantum network. Our results can be summarized as follows. Suppose a quantum network is described by an undirected graph $G$ with one source node and two (three) target nodes, and each quantum channel on the quantum network $G$ can transmit one $q$-dimensional quantum system in a single session. Furthermore, suppose that there exists a classical solvable multicast network code with source rate $r$ for a classical network described by an acyclic directed graph $G'$, where $G$ is an undirected underlying graph of $G'$. We showed that under the above assumptions, our protocol can multicast $1 \to 2$ ($1 \to 3$) asymmetric optimal UQCs of a $q^r$-dimensional state from the source node to

the target nodes by consuming a small amount of entanglement that does not scale with $q$, which is shared among the target nodes. We further showed that our protocol can be used for efficient preparation of quantum telecloning over a quantum network.

As we have discussed in the previous subsection, when we use a known scheme of a $1 \rightarrow N$ asymmetric optimal UQC for $N \geq 4$, a protocol which derived straightforward extension of our scheme need $O(\log d)$ ebits shared among target nodes. Thus, the extension of our protocol for $1 \rightarrow n$ asymmetric optimal UQCs for $n \leq 4$ is not so straightforward. Hence, we leave this study as our future work.

In this paper, we assumed that all quantum channels on a quantum network are noiseless. This assumption can be justified by considering our protocol as a protocol on the layer on which error correction has been already implemented. However, as is well known, such complete error correction of a quantum channel is beyond the present technology. For example, the long-distance distribution of GHZ-type states, which is nothing but the purpose of the protocol of Kobayashi et al., is a huge challenge in quantum networks. We note that as a practical protocol for this purpose, recently, a protocol to distribute the postselected GHZ state by using entanglement swapping is proposed [106].

As we have mentioned above, in this paper, we assume that error correction of quantum channels has been applied on a quantum network before our protocol is implemented. In other words, we assume that optimization is applied separately on these two layers. On the other hand, if we optimize these two layers simultaneously, we may derive a better protocol. We leave this study as future work.

The protocol of Kobayashi et al., which is one of the main subroutines of our protocol, gives an efficient protocol to share GHZ states. On the other hand, as we have explained in Section 1, except GHZ-states, there are many classes of multipartite entangled states that are incomparable to the class of GHZ states under SLOCC-like W states, Dicke states (generalized W states), etc. It may be desirable to find an efficient quantum network protocol to share these classes of states. This may be also our future work.

**Author Contributions:** Conceptualization, M.O.; methodology, M.O.; validation, M.O.; investigation, Y.H. and M.O.; writing—original draft preparation, Y.H.; writing—review and editing, M.O.; visualization, Y.H. and M.O.; supervision, M.O.; project administration, M.O.; funding acquisition, M.O. All authors have read and agreed to the published version of the manuscript.

**Funding:** This research was funded by the JSPS Kakenhi (C) No. 16K00014, No. 17K05591, No. 20K03779, and No. 21K03388.

**Institutional Review Board Statement:** Not applicable.

**Data Availability Statement:** Not applicable.

**Conflicts of Interest:** The authors declare no conflict of interest.

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
