# Peer review of "Asymmetric Quantum Multicast Network Coding: Asymmetric Optimal Cloning over Quantum Networks"

_applsci, doi:10.3390/app12126163_

Round 1
Reviewer 1 Report
Manuscript 1695071 considers a quantum version of multicast network coding as a multicast protocol for sending universal quantum clones (UQCs) from a source node to target nodes on a quantum network. The authors derive protocols for multicasting 1—2 and 1—3 asymmetric UQCs of a qr-dimensional state to two and three target nodes, respectively. The results of this manuscript may have some interesting. As far as I am concerned, this manuscript cannot be published in applied science unless the authors have addressed the following issues.
1. The main issue in this manuscript is that did not have enough physical explanation and definition. What do symmetric and asymmetric universal quantum clones mean and what's the difference between them?
2. The results of this manuscript are not quantitatively analyzed. What is the fidelity for the multicasting 1—2 and 1—3 asymmetric UQCs of a qr-dimensional state to two and three target nodes? The authors should add the corresponding figures to show corresponding results.
3. What is the advantage of this manuscript result without a reasonable comparison with the existing references.
4. Using quantum technologies, the quantum advantage of machine learning has been demonstrated [r1]. To add the readability, this article should be added in the sentences “stand-alone quantum computers are expected to show innovative performance in various fields, such as machine learning [5-8].”
[r1] Experimental Quantum Advantage with Quantum Coupon Collector, Research 2022, 9798679 (2022).
5. The long-distance distribution of GHZ-type states is a huge challenge in quantum networks. The post-selected GHZ state can solve the distribution issue by using entanglement swapping [r2], which also helps this manuscript to share GHZ-type states among the target nodes. I suggest that the authors cite it and make a discussion.
[r2] Long-Distance Measurement-Device-Independent Multiparty Quantum Communication, PRL 114, 090501 (2015).
Reviewer 2 Report
Dear Editor,
In this manuscript, the authors claim to be extending Owari’s previous results symmetric optimal cloning over quantum networks to asymmetric optimal cloning. The manuscript is clear and well-written in general and calculations are easy to follow. Considering its significance and potential to contribute to the field, it can be considered for publication in Applied Sciences. However, I would like to raise a few concerns and suggestions.
Note that I chose “English language and style are fine/minor spell check required” because it was the closest choice but I think there is not much problem in the language and writing of the manuscript.
1- The extended works (major Refs.[38-40]) are patents in Japanese, that I could not access and read. This is problematic. It would be better to also cite the English versions of those works. Though there is some explanation in the manuscript, it would not be sufficient.
2- Ref.[39] and Ref.[40] look like the same, only patent number and years are different, but the titles are the same. Is there a type there?
3- While searching for the contents of the patents, I came across a recent work with almost the same title: https://iopscience.iop.org/article/10.1088/1674-1056/ac20c6 Although the work done is different, because of the same title, mentioning that work and pointing out the differences would be good.
4- The general assumption that the protocol is implemented on the layer free from errors is reasonable, allowing to consider noise-free channels/networks. However, considering more general scenarios and architectures with alternative layering, optimal cloning over noisy networks might lead to interesting results and protocols. I can suggest the authors to add at least a short discussion on that for opening new insights for future research.
5- A similar argument can be made for the class of multipartite-entangled states. I would like to see at least a short discussion on future research considering other major types of states, such as W state or general Dicke states.
Reviewer 3 Report
In this manuscript, the authors proposed a quantum multicast protocol for sending universal quantum clones from the source node to target nodes on a quantum network. The manuscript is well written, and the results are presented in a convincible and comprehensive way. The results derived in this work, including the new protocol of multicasting asymmetric optimal universal quantum clones and the relation between classical and quantum multicast network code, would be useful for the future studies of quantum communication and quantum internet . I think this manuscript would be suitable for publication on Appl. Sci. after some minor typo errors being corrected, such as in line 483, “n<=4” should be “n>=4”.
Reviewer 4 Report
This manuscript introduced a protocol to multicast asymmetric optimal UQCs from one source node to two (three) target nodes over a quantum network effectively. The proposed protocol is developed basing on the previous research from Owari and Kobayashi to achieve the asymmetric optimal universal cloning and multicast of a quantum state, respectively.
Overall, the quality of this manuscript is good. The authors gave a good description on the running procedure of the protocol with different scenarios, and the correctness of asymmetric optimal clones after multicasting is proved as well.
Some shortcomings, however, still exist in the manuscript.
- the quality of abstract should be improved. The definition of the research problem is not accurate enough, and the proposed method to be adopted is not clear too.
- It is suggested that the importance of the research problem and background should be given more in detail in the Introduction.
- The proportion of cited references publishing in recent 3-5 years is relatively small.
- in Line 342-343, the cited equations should be “27”, and the “Eq. (39)” should be replaced by “Eq. (29)” in Line 344.
Reviewer 5 Report
The authors investigate distributed quantum cloning protocol, where quantum information is transmitted over a specific quantum network to produce asymmetric quantum clones at the target nodes of the network. It is assumed that the network has a limited transmission capacity, which makes the studied problem non-trivial. The authors build on previous results of Kobayashi, et al. It should be emphasized that while the problem studied in the present manuscript is related to the work of Kobayashi, it is a different and novel task that deserves to be investigated. The authors show that the optimal asymmetric cloning over the quantum network is possible within the considered setting if the target nodes share a limited amount of entanglement. In brief, the asymmetric cloning is first implemented at the source and the ancilla is measured in the computational basis. The state of the clones is then mapped to a single qudit and distributed through the network, which yields an entangled state at the output. This entangled state is then deterministically converted to the correct state of the clones using the result of measurement on the ancilla as a classical control information.
The protocol is interesting and it provides new insights into the ways we can manipulate and transmit quantum information over quantum networks. The manuscript is well and clearly written (taking into account the unavoidably highly technical content of the work) and I recommend its publication.
The authors may consider citing the paper [J. Fiurasek, R. Filip and N. J. Cerf, Quantum Information & Computation 5, 583–592 (2005)] that provides some additional details on asymmetric quantum cloners.
Could there be some relation between the present protocol and the quantum telecloning where one combines quantum teleportation and cloning?
Round 2
Reviewer 1 Report
The authors have addressed all concerns raised by me. However, I recommend the authors to check the article carefully, there are still some minor typos. For example, the reference [r2] has an error on line 625, which is should be in the reference list [Phys. Rev. Lett. 114, 090501 (2015)].
